# Scalable Cross-View Sample Alignment for Multi-View Clustering with View Structure Similarity

Jun Wang[1]    Zhenglai Li[2]    Chang Tang[3]    Suyuan Liu[1]    Hao Yu[1]    Chuan Tang[1]
Miaomiao Li[4*]    Xinwang Liu[1*]

[1]National University of Defense Technology, Changsha, China
[2]Shenzhen Institutes of Advanced Technology, Shenzhen, China
[3]Huazhong University of Science and Technology, Wuhan, China
[4]Changsha College, Changsha, China

## Abstract

Most existing multi-view clustering methods aim to generate a consensus partition across all views, based on the assumption that all views share the same sample arrangement. However, in real-world scenarios, the collected data across different views is often unsynchronized, making it difficult to ensure consistent sample correspondence between views. To address this issue, we propose a scalable sample-alignment-based multi-view clustering method, referred to as SSA-MVC. Specifically, we first employ a cluster-label matching (CLM) algorithm to select the view whose clustering labels best match those of the others as the benchmark view. Then, for each of the remaining views, we construct representations of non-aligned samples by computing their similarities with aligned samples. Based on these representations, we build a similarity graph between the non-aligned samples of each view and those in the benchmark view, which serves as the alignment criterion. This alignment criterion is then integrated into a late-fusion framework to enable clustering without requiring aligned samples. Notably, the learned sample alignment matrix can be used to enhance existing multi-view clustering methods in scenarios where sample correspondence is unavailable. The effectiveness of the proposed SSA-MVC algorithm is validated through extensive experiments conducted on eight real-world multi-view datasets.

## 1  Introduction

Clustering aims to assign each sample to its corresponding class by leveraging the intrinsic similarities within the original data [1]. With the rapid advancement of science and technology, data have become increasingly diverse in their forms of representation. The same object can often be described from multiple perspectives. For instance, video content can be represented through audio, visual, and textual modalities. Such heterogeneous but complementary data representations are collectively referred to as multi-view data [2, 3, 4, 5]. To fully exploit the rich semantic information embedded in multi-view data, a variety of advanced multi-view clustering algorithms have been developed in recent years [6, 7, 8, 9]. These methods have demonstrated promising performance across a wide range of real-world applications by effectively integrating complementary information from multiple views.

Despite the effectiveness of these approaches in integrating multi-view information, they typically rely on the assumption of strict one-to-one correspondence among samples across different views, which is an idealized condition in practical applications [10, 11, 12]. In real-world scenarios, variations in

---

*Corresponding Author

39th Conference on Neural Information Processing Systems (NeurIPS 2025).

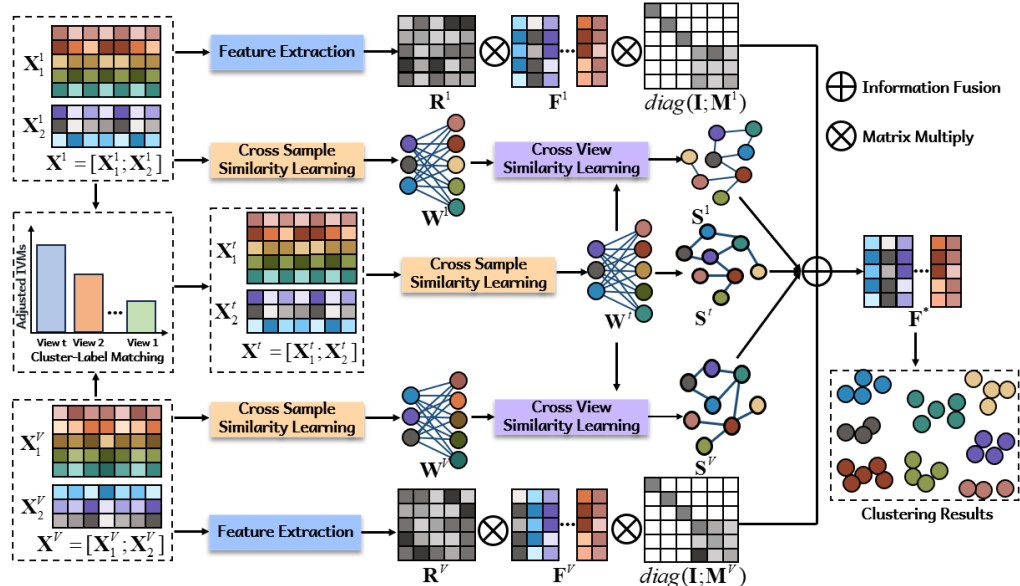

Figure 1: The flowchart of the proposed method. First, the baseline view is selected based on the CLM criterion. Next, feature representations of the unaligned samples, denoted as $\{\mathbf{W}^v\}_{v=1}^V$, are constructed. Subsequently, the cross-view similarity graphs $\{\mathbf{S}^v\}_{v=1}^V$ between the baseline view and the other views are established. Finally, these cross-view similarity graphs serve as alignment constraints within a late fusion multi-view clustering framework to obtain a unified partition matrix $\mathbf{F}^*$. The final clustering results are then derived by applying $k$-means clustering on $\mathbf{F}^*$.

sample organization or ordering across views commonly lead to inconsistent or misaligned sample correspondences. To this end, some studies attempt to achieve sample alignment jointly with the learning of data representations. Given the effectiveness of the Hungarian algorithm in assignment problems [13], Huang et al. [14] integrated it into their clustering framework to facilitate sample alignment. However, due to semantic discrepancies between views and high intra-class similarity within views, establishing strict one-to-one alignment based solely on sample similarity limits tolerance to noise and misalignment. To address this, Yang et al. [15] proposed alignment at the class level, reformulating it as a class identification problem and introducing a noise-robust contrastive loss to improve robustness. Furthermore, Ren et al. [16] leveraged sample commonality and view diversity to adaptively construct alignment matrices and designed an unsupervised data completion mechanism to handle incomplete or unaligned data.

Although the aforementioned algorithms have shown promising performance in multi-view clustering with sample alignment, they still face several limitations. (1) Due to semantic discrepancies across views and the absence of supervision, establishing strict one-to-one correspondences is often difficult. In real-world scenarios, the sample relationships of different views are typically many-to-many, and enforcing strict matching may introduce noise and lead to sub-optimal alignment [17]. (2) While some recent methods employ joint learning frameworks that integrate alignment with feature representation to enhance performance, they often fail to model explicit alignment relationships, thus limiting their scalability to other multi-view clustering methods that are not applicable in sample non-alignment scenarios [18, 19]. (3) Clustering performance is heavily influenced by the choice of a benchmark view, yet selecting an appropriate one remains an open challenge in current approaches [20].

Therefore, we propose a scalable multi-view clustering algorithm that integrates sample alignment into a unified clustering framework. To mitigate the impact of structurally noisy or disordered views, we first employ the CLM algorithm [21] to select the view that exhibits the highest structural consistency with the underlying semantic labels, designating it as the baseline. Considering that samples within the same subspace can be linearly reconstructed by their peers [22], we reformulate the alignment task as a similarity-based reconstruction problem rather than relying on rigid one-to-one index matching. Specifically, each view is structurally characterized by computing the similarity between unaligned samples and the rest of the view. Then, an alignment relationship is established by comparing these

structural representations to that of the baseline view. Finally, the resulting alignment matrices are incorporated into a late fusion clustering framework, enabling effective alignment without the need for direct correspondence. Furthermore, the learned alignment relationship can be reused as auxiliary information to enhance the performance of existing multi-view clustering methods under misaligned conditions.

Overall, the main contributions of this paper are listed as follows:

- We propose to select the baseline view by measuring the similarity between sample cluster distributions and their corresponding labels within each view, effectively minimizing the impact of irrelevant or noisy structural information on the alignment process.

- We propose a structural representation for each view based on the correlation between non-aligned and aligned samples. This representation guides cross-view alignment by integrating sample-level features with intrinsic structural information.

- An alternating optimization algorithm is proposed to efficiently solve the model. Its effectiveness is validated through extensive experiments on eight multi-view datasets.

## 2 Preliminaries

### 2.1 Adaptive Neighbor Graph Learning

Graph-based multi-view clustering algorithms have attracted considerable attention in recent years due to their strong capability in capturing the intrinsic structural information embedded in the original data [23, 24, 25, 26]. Based on the fact that samples within the same cluster or samples with smaller pairwise distances tend to exhibit higher similarity than those from different clusters, Nie et al. [27] proposed a clustering algorithm that constructs a nearest-neighbor graph to capture local structural relationships. The objective function of this method is formulated as follows:

$$\min_{\mathbf{S}} \sum_{i=1}^{n} \sum_{j=1}^{n} \left( \|\mathbf{x}_i - \mathbf{x}_j\|_2^2 \cdot s_{ij} + \beta s_{ij}^2 \right) \quad s.t. \ \mathbf{s}_i^\top \mathbf{1} = 1, 0 \leq s_{ij} \leq 1, \tag{1}$$

where $\mathbf{x}_i$ and $\mathbf{x}_j$ denotes the $i$-th and $j$-th samples of the original data matrix $\mathbf{X} \in \mathbb{R}^{n \times d}$, where $n$ is the total number of samples and $d$ is the feature dimension. The variable $s_{ij}$ indicates the similarity between samples $\mathbf{x}_i$ and $\mathbf{x}_j$. The parameter $\beta$ is a regularization coefficient that balances the trade-off between the similarity graph learning and the sparsity of the graph, and it can be adaptively tuned during the optimization process. For a more detailed description of Eq. (1), please refer to [27].

### 2.2 Late Fusion based Multi-view Clustering

Given multi-view datasets $\mathbf{X}^v \in \mathbb{R}^{n \times d_v}$, where $d_v$ denotes the feature dimensionality of the $v$-th view, late fusion-based multi-view clustering methods aim to extract partition-level clustering information from each view. A unified partition matrix is then obtained by integrating the partition information from all views. Specifically, assuming that the base partition matrices $\mathbf{F}^v \in \mathbb{R}^{n \times d_u}$ are obtained from $\mathbf{X}^v$ via eigen-decomposition or other representation learning techniques, where $d_u$ denotes the latent feature dimension, the typical mathematical formulation can be expressed as follows [28]:

$$\max_{\mathbf{F}^*} \Phi(\mathbf{F}^*, \mathbf{F}^v) + \lambda \, \Psi(\mathbf{F}^*), \tag{2}$$

where $\Phi(\cdot)$ denotes the partition fusion module, which integrates the base partitions into a unified one, and $\Psi(\cdot)$ represents a regularization term designed to preserve desirable properties such as smoothness [29], sparsity [30], or consistency across views [31, 32].

## 3 Proposed Method

### 3.1 Cross Sample Similarity Learning

Late fusion-based multi-view clustering algorithms have attracted substantial attention due to their demonstrated effectiveness, and a variety of advanced methods have been proposed within this

framework [33, 34, 35]. However, a common assumption in these algorithms is the existence of a strict one-to-one correspondence between samples across all views during the fusion of partition information. In practice, this assumption is often idealized. Due to temporal misalignment during data acquisition or storage constraints, mismatches between samples across different views frequently occur. Under such circumstances, directly fusion partition-level information without addressing the cross-view sample alignment may introduce irrelevant or inconsistent structural information, thereby degrading the quality of the unified partition and affecting the final clustering performance.

To overcome the limitations of traditional late fusion strategies, several algorithms have introduced a sample alignment matrix jointly with view-specific representation learning, integrating the two processes into a unified framework to enable mutual reinforcement [36, 37, 38]. In the context of unsupervised learning, sample alignment is typically inferred by exploiting the intrinsic feature similarities among data samples. However, due to the presence of substantial cross-view heterogeneity, the assumption of semantic consistency across views is often difficult to capture using rigid one-to-one matching strategies. Such hard alignment approaches fail to model potential one-to-many or many-to-one semantic relationships across views, thereby overlooking alternative and potentially meaningful alignment relationships. Moreover, although some methods attempt to embed the sample alignment process into neural network-based representation learning, they often do not explicitly model the alignment relationships themselves, which limits the scalability of these approaches.

In light of the above challenges, we propose a novel strategy that utilizes the correlation between unaligned and aligned samples within each view as a view-specific structural representation. Specifically, given an unaligned multi-view dataset $\mathbf{X}^v = [\mathbf{X}_1^v; \mathbf{X}_2^v]$, where $\mathbf{X}_1^v \in \mathbb{R}^{n_1 \times d_v}$ and $\mathbf{X}_2^v \in \mathbb{R}^{n_2 \times d_v}$ denote the aligned and unaligned samples in the $v$-th view, respectively, and where $n_1$ and $n_2$ are the corresponding numbers of aligned and unaligned samples, we construct the structural representations of the unaligned samples for each view based on Eq. (1), i.e.,

$$\min_{\{\mathbf{W}^v\}_{v=1}^V} \sum_{v=1}^V \sum_{i=1}^{n_2} \sum_{j=1}^{n_1} \left\| \mathbf{X}_{1[i,:]}^v - \mathbf{X}_{2[j,:]}^v \right\|_2^2 w_{ij}^v + \beta \left( w_{ij}^v \right)^2 \quad s.t. \ \mathbf{w}_i^{v\top} \mathbf{1} = 1, 0 \le w_{ij}^v \le 1, \quad (3)$$

where $V$ denotes the total number of views. The matrix $\mathbf{W}^v \in \mathbb{R}^{n_2 \times n_1}$ represents the constructed feature representation for the $v$-th view in the presence of sample non-alignment. It is worth noting that the above construction of the feature representation for each view is not unique. Here we adopt the formulation given in Eq. (1) for simplicity. Nevertheless, alternative learning mechanisms could also be employed within our framework.

### 3.2 Cross View Similarity Learning

After obtaining the feature representations for all views, a key challenge lies in constructing a reliable sample alignment across views. A straightforward approach is to randomly select one view as the baseline and align the remaining views to it. However, due to inevitable noise introduced during data collection, some views may contain structural information that does not reflect the true underlying cluster distribution. To mitigate the impact of such irrelevant or misleading information on the alignment process, we adopt the CLM algorithm to identify the most reliable baseline view. Specifically, the view that exhibits the highest consistency between its sample distribution structure and the semantic labels is selected as the baseline. The detailed selection process is defined as:

$$H(Y, \mathbf{X}, d^2) = \frac{\exp\left( \frac{1}{\sigma_{d^2} n} \sum_{\mathbf{x} \in \mathbf{X}} d^2(\mathbf{x}, y) \right)}{\exp\left( \frac{1}{\sigma_{d^2} n} \sum_{i=1}^k \sum_{\mathbf{x} \in Y_i} d^2(\mathbf{x}, y_i) \right)} \times \frac{\sum_{i=1}^k |Y_i| d^2(y_i, y)}{\sigma_{d^2} n (k-1)} \quad (4)$$

$$CLM(\mathbf{X}) = \frac{1}{2\binom{k}{2}} \sum_{\substack{G \subseteq Y \\ |G|=2}} \frac{1}{1 + \exp\left( -\delta \cdot H(G, \mathbf{X}, d^2) \right)} \quad (5)$$

where $Y = \{Y_1, Y_2, \cdots, Y_k\}$ denotes the ground-truth cluster assignment of the dataset $\mathbf{X}$, and $k$ is the total number of clusters. Let $y_i = \overline{Y_i}$ denote the mean of the samples in the $i$-th cluster, and $c = \overline{\mathbf{X}}$ denote the mean of all samples. The function $d^2(\cdot)$ represents the squared Euclidean distance, and $\sigma_{d^2} = std(d^2(x, c) | \mathbf{x} \in \mathbf{X})$ denotes the standard deviation of the distances between the original data samples and the global centroid. The parameter $\delta$ is a pre-defined scaling factor. Based on

the above formulation, we compute a matching score that quantifies the consistency between the structural distribution of samples and the corresponding semantic clusters in each view. The view with the highest matching score is then selected as the baseline. Accordingly, the cross-view structural similarity graph $\mathbf{S}^v$ is constructed as follows:

$$\min_{\mathbf{S}^v} \sum_{\substack{v=1 \\ v \neq t}}^{V} \|\mathbf{w}_i^t - \mathbf{w}_j^v\|_2^2 s_{ij}^v + \beta(s_{ij}^v)^2 \quad s.t.\ t = \arg\max_v CLM(\mathbf{X}^v), \mathbf{s}_i^\top \mathbf{1} = 1, 0 \leq s_{ij}^v \leq 1, \quad (6)$$

where $\mathbf{S}^v$ denotes the similarity graph that captures the structural correspondence between the unaligned samples in the $v$-th view and those in the baseline view, denoted by $t$. As a special case, when $v = t$, we define the similarity graph as the identity matrix, i.e., $\mathbf{S}^t = \mathbf{I}$.

### 3.3 Sample-Aligned Late Fusion Strategy

In general, a higher similarity between samples implies a greater likelihood of a semantic match. Based on this intuition, we propose to capture the matching criterion between unaligned samples across views by leveraging cross-view sample similarity. As discussed earlier, the ideal scenario assumes a strict one-to-one correspondence between samples across views. However, in practice, such hard 0-1 alignments are difficult to establish in the absence of external supervision, due to the presence of noise and structurally irrelevant information in certain views.

To address this challenge, we reformulate the alignment problem by reconstructing unaligned samples using samples within their corresponding subspace, rather than explicitly matching index positions across views. In this way, the alignment is achieved in a soft and structure-preserving manner. By integrating this alignment strategy with the late fusion-based multi-view clustering framework, we formulate the final objective function as follows:

$$\max_{\mathbf{R}^v, \mathbf{F}^*, \mathbf{M}^v, \alpha_v} \mathrm{Tr}\left(\mathbf{F}^{*\top}\left(\alpha_t \mathbf{F}^t \mathbf{R}^t + \sum_{\substack{v=1 \\ v \neq t}}^{V} \alpha_v \begin{bmatrix} \mathbf{I} & \mathbf{0} \\ \mathbf{0} & \mathbf{M}^v \end{bmatrix} \mathbf{F}^v \mathbf{R}^v\right)\right) + \lambda \sum_{v=1}^{V} \mathrm{Tr}(\mathbf{M}^{v\top} \mathbf{S}^v)$$

$$s.t.\ t = \arg\max_v CLM(\mathbf{X}^v), \mathbf{F}^{*\top}\mathbf{F}^* = \mathbf{I}, \mathbf{R}^{v\top}\mathbf{R}^v = \mathbf{I}, \sum_{v=1}^{V} \alpha_v^2 = 1, \mathbf{M}^{v\top}\mathbf{M}^v = \mathbf{I}, \quad (7)$$

where $\mathbf{M}^v$ denotes the sample realignment matrix that maps the unaligned samples in the $v$-th view to those in the baseline view, while $\mathbf{R}^v$ represents the feature rotation matrix used to align the feature space. The scalar $\alpha_v$ indicates the weight assigned to the $v$-th view, and $\lambda$ is a hyperparameter that controls the trade-off between feature information and structural information.

## 4 Optimization

### 4.1 Optimization Algorithms

In this section, we develop an iterative optimization algorithm to solve the objective function in Eq. (7) with respect to the variables $\mathbf{R}^v$, $\mathbf{F}^*$, $\mathbf{M}^v$, and $\alpha_v$. The detailed optimization procedure is described as follows:

**Update $\mathbf{F}^*$**: When optimizing $\mathbf{F}^*$ while keeping all other variables fixed, the objective function in Eq. (7) can be equivalently reformulated as:

$$\max_{\mathbf{F}^*} \sum_{v=1}^{V} \mathrm{Tr}\left(\alpha_v \mathbf{F}_1^{*\top} \mathbf{F}_1^v \mathbf{R}^v + \alpha_v \mathbf{F}_2^{*\top} \mathbf{M}^v \mathbf{F}_2^v \mathbf{R}^v\right) \quad s.t.\ \mathbf{F}^{*\top}\mathbf{F}^* = \mathbf{I}, \mathbf{F}^* = \begin{bmatrix} \mathbf{F}_1^* \\ \mathbf{F}_2^* \end{bmatrix}. \quad (8)$$

Since $\mathbf{F}_1^*$ and $\mathbf{F}_2^*$ are independent of each other, they can be optimized separately to obtain the complete solution for $\mathbf{F}^*$. Specifically, when optimizing the variable $\mathbf{F}_1^*$, the objective function in Eq. (8) can be equivalently rewritten as:

$$\max_{\mathbf{F}_1^*} \sum_{v=1}^{V} \mathrm{Tr}(\mathbf{F}_1^{*\top} \alpha_v \mathbf{F}_1^v \mathbf{R}^v) \quad s.t.\ \mathbf{F}_1^{*\top}\mathbf{F}_1^* = \mathbf{I}. \quad (9)$$

The optimal solution to Eq. (9) can be obtained by performing singular value decomposition (SVD) on the matrix $\alpha_v \mathbf{F}_1^v \mathbf{R}^v$. Since the optimization of $\mathbf{F}_2^v$ follows a procedure analogous to that of $\mathbf{F}_1^v$, we omit the details here for brevity. Once the optimal solutions for both $\mathbf{F}_1^v$ and $\mathbf{F}_2^v$ are obtained, the final solution for $\mathbf{F}^*$ is constructed by concatenating the two parts.

**Update $\mathbf{R}^v$**: When all other variables are fixed, the optimization of $\mathbf{R}^v$ in Eq. (7) can be equivalently reformulated as:

$$\max_{\mathbf{R}^v} \alpha_v \operatorname{Tr}\left( \mathbf{R}^{v\top} \mathbf{F}^{v\top} \begin{bmatrix} \mathbf{I} & \mathbf{0} \\ \mathbf{0} & \mathbf{M}^v \end{bmatrix} \mathbf{F}^* \right) \quad s.t.\ \mathbf{R}^{v\top} \mathbf{R}^v = \mathbf{I}. \tag{10}$$

Let $\mathbf{Q}^v = \alpha_v \mathbf{F}^{v\top} \begin{bmatrix} \mathbf{I} & 0 \\ 0 & \mathbf{M}^v \end{bmatrix} \mathbf{F}^*$, the optimal solution for the variable $\mathbf{R}^v$ can then be obtained like that of $\mathbf{F}_1^*$, specifically by performing singular value decomposition on $\mathbf{Q}^v$.

**Update $\alpha_v$**: By fixing other variables, the Eq. (7) can be formulated as:

$$\max_{\boldsymbol{\alpha}} \sum_{v=1}^{V} \alpha_v \gamma_v \quad s.t.\ \sum_{v=1}^{V} \alpha_v^2 = 1, \tag{11}$$

where $\gamma_v = \operatorname{Tr}(\mathbf{F}^{*\top} \mathbf{C}^v \mathbf{F}^v \mathbf{R}^v)$. According to the Cauchy inequality, the optimal solution to the above optimization problem can be derived in closed form as:

$$\alpha_v = \frac{\gamma_v}{\sqrt{\sum_{v=1}^{V} \gamma_v^2}}. \tag{12}$$

**Update $\mathbf{M}^v$**: By fixing $\mathbf{F}^*$, $\mathbf{R}^v$, and $\alpha_v$, $\mathbf{M}^v$ can be optimized by solving the following subproblem:

$$\max_{\mathbf{M}^v} \alpha_v \operatorname{Tr}\left( \mathbf{M}^{v\top} \mathbf{F}_2^* \mathbf{R}^{v\top} \mathbf{F}_2^{v\top} \right) + \lambda \operatorname{Tr}\left( \mathbf{M}^{v\top} \mathbf{S}^v \right) \quad s.t.\ \mathbf{M}^{v\top} \mathbf{M}^v = \mathbf{I}. \tag{13}$$

Let $\mathbf{T} = \alpha_v \mathbf{F}_2^* \mathbf{R}^{v\top} \mathbf{F}_2^{v\top} + \lambda \mathbf{S}^v$. Following an optimization procedure similar to that for the variable $\mathbf{M}^v$, the optimal solution can be obtained by performing SVD on the matrix $\mathbf{T}$.

In summary, the detailed procedure of the proposed method is described in the Appendix A.2.

## 4.2 Convergence Property

In the above optimization process, each subproblem is independent, and its corresponding optimal solution can be obtained. Consequently, the proposed algorithm converges within a few iterations according to Theorem 1. A detailed convergence proof is provided in the Appendix A.3.

**Theorem 1.** *The proposed optimization algorithm is guaranteed to converge to a local optimum of the SSA-MVC method.*

## 4.3 Computational Complexity Analysis

In the proposed method, the primary computational complexity arises from three components: cross-sample similarity learning, cross-view similarity learning, and sample-aligned late fusion. Specifically, the computational cost for obtaining the feature representations $\{\mathbf{W}^v\}_{v=1}^{V} \in \mathbb{R}^{n_2 \times n_1}$ is $\mathcal{O}(V n_2 K d_{max})$, where $K$ denotes the number of neighbors and $d_{max} = \max\{d_1, d_2, \cdots, d_V\}$ represents the maximum feature dimension across all views. For the CLM algorithm, the complexity is $\mathcal{O}(n d_{max})$, while the construction of the cross-view similarity graph requires $\mathcal{O}(V n_2 n_1 K)$ operations. Finally, the computational cost of the late fusion step is $\mathcal{O}(n^2)$, mainly due to the generation of the partition matrix $\mathbf{F}^v$. Consequently, the overall computational complexity is $\mathcal{O}(n^2)$.

# 5 Experiments

## 5.1 Datasets

To further validate the effectiveness of the proposed method, we conduct experiments on eight real-world multi-view datasets, including Yale, 3sources, MSRCV, 100leaves, HW, Scene, EMNIST, and Hdigit. The detailed summary of them is provided in Appendix A.4.

Table 1: ACC comparison of all methods with and without Hungarian alignment on eight multi-view datasets under a sample alignment ratio $\rho = 50\%$.

| Method | Yale | 3sources | MSRCV | 100leaves | HW | Scene | EMNIST | Hdigit |
|---|---|---|---|---|---|---|---|---|
| EEOMVC | 58.18±0.00 | 59.76±0.00 | 72.86±0.00 | 65.44±0.00 | 93.85±0.00 | 26.91±0.00 | 46.11±0.00 | 65.93±0.00 |
| EEOMVC + Hungarian | 52.73±0.00 | 48.52±0.00 | 71.90±0.00 | 67.19±0.00 | 67.30±0.00 | 26.00±0.00 | 44.24±0.00 | 62.28±0.00 |
| DealMVC | 33.94±0.00 | 31.01±0.00 | 28.10±0.00 | 7.69±0.00 | 47.94±0.00 | 22.54±0.00 | 45.62±0.00 | 65.66±0.00 |
| DealMVC + Hungarian | 24.85±0.00 | 29.11±0.00 | 28.29±0.00 | 9.34±0.00 | 39.97±0.00 | 21.86±0.00 | 37.83±0.00 | 82.72±0.98 |
| MVCAN | 32.48±2.50 | 31.12±2.31 | 58.14±1.76 | 49.51±1.24 | 50.62±0.42 | 33.30±0.41 | 45.06±1.02 | 65.50±2.98 |
| MVCAN + Hungarian | 33.09±1.56 | 49.37±4.60 | 51.20±2.80 | 40.71±1.53 | 47.08±3.56 | 29.75±0.54 | 49.56±9.49 | 57.37±3.76 |
| EBMGC | 39.39±0.00 | 38.46±0.00 | 42.86±0.00 | 33.94±0.00 | 56.75±0.00 | 21.58±0.00 | 33.50±0.00 | 50.91±0.00 |
| EBMGC + Hungarian | 32.73±0.00 | 40.24±0.00 | 47.14±0.00 | 33.94±0.00 | 51.70±0.00 | 26.33±0.00 | 41.00±0.00 | 59.46±0.00 |
| Vsc_mH | 53.94±0.00 | 62.13±0.00 | 64.29±0.00 | 38.56±0.00 | 42.50±0.00 | 28.03±0.00 | 46.47±0.00 | 65.19±0.00 |
| OpVuC | 53.94±0.00 | 57.40±0.00 | 30.00±0.00 | 53.13±0.00 | 30.10±0.00 | 31.82±0.00 | 51.09±0.00 | 62.90±0.00 |
| DCMVC | 27.15±1.01 | 46.51±4.52 | 45.71±2.29 | 48.83±0.83 | 69.34±1.01 | 26.02±0.60 | 59.55±3.60 | 65.74±2.31 |
| DCMVC + Hungarian | 23.88±1.19 | 35.15±1.10 | 44.57±2.46 | 39.34±0.95 | 50.47±0.99 | 24.07±0.45 | 40.11±1.28 | 35.80±0.99 |
| LMTC | 52.58±3.76 | 48.28±3.49 | 53.29±3.15 | 35.58±0.94 | 64.99±1.16 | 28.96±0.92 | 41.91±0.80 | 59.25±0.30 |
| LMTC + Hungarian | 54.61±4.74 | 48.05±1.18 | 55.43±3.29 | 35.30±1.45 | 54.24±2.72 | 28.53±0.86 | 41.69±0.70 | 55.46±2.13 |
| TMSL | 24.82±1.70 | 42.25±2.66 | 43.98±0.97 | 47.47±1.40 | 62.61±0.61 | 29.13±0.49 | OOM | OOM |
| TMSL + Hungarian | 68.79±2.78 | 56.21±1.05 | 44.45±1.45 | 47.12±1.11 | 53.12±0.05 | 27.02±0.24 | OOM | OOM |
| DSTL | 35.91±1.93 | 61.54±0.00 | 39.48±3.81 | 36.87±1.42 | 47.61±1.57 | 20.45±0.59 | 28.87±0.40 | 40.90±0.68 |
| DSTL + Hungarian | 37.73±2.76 | 59.76±0.19 | 43.71±1.43 | 30.57±1.03 | 43.33±0.51 | 19.31±0.79 | 30.36±0.33 | 50.09±0.00 |
| Ours | 64.24±3.62 | 64.44±1.29 | 83.52±0.39 | 70.63±1.29 | 96.55±0.00 | 35.91±0.27 | 77.38±3.14 | 71.78±1.26 |

Table 2: NMI comparison of all methods with and without Hungarian alignment on eight multi-view datasets under a sample alignment ratio $\rho = 50\%$.

| Method | Yale | 3sources | MSRCV | 100leaves | HW | Scene | EMNIST | Hdigit |
|---|---|---|---|---|---|---|---|---|
| EEOMVC | 62.23±0.00 | 39.32±0.00 | 56.08±0.00 | 75.62±0.00 | 88.20±0.00 | 16.59±0.00 | 32.54±0.00 | 70.96±0.00 |
| EEOMVC + Hungarian | 57.37±0.00 | 31.87±0.00 | 56.93±0.00 | 77.03±0.00 | 62.65±0.00 | 18.26±0.00 | 29.18±0.00 | 53.11±0.00 |
| DealMVC | 38.08±0.00 | 6.69±0.74 | 14.00±3.92 | 25.34±0.44 | 27.20±0.89 | 11.89±1.09 | 31.39±0.59 | 39.90±1.52 |
| DealMVC + Hungarian | 23.65±0.00 | 7.31±0.48 | 13.08±0.17 | 27.34±3.63 | 26.08±2.49 | 16.31±3.23 | 22.80±0.48 | 65.46±1.81 |
| MVCAN | 38.43±1.87 | 12.87±1.67 | 46.19±1.94 | 69.50±0.95 | 32.31±0.48 | 30.96±0.89 | 20.14±0.13 | 60.89±1.68 |
| MVCAN + Hungarian | 38.53±1.22 | 47.72±3.00 | 35.16±4.14 | 62.08±1.45 | 45.05±5.03 | 25.76±0.33 | 37.94±15.14 | 53.82±3.54 |
| EBMGC | 43.31±0.00 | 23.68±0.00 | 22.79±0.00 | 58.22±0.00 | 35.99±0.00 | 11.14±0.00 | 17.99±0.00 | 25.07±0.00 |
| EBMGC + Hungarian | 38.18±0.00 | 23.98±0.00 | 24.79±0.00 | 58.22±0.00 | 29.52±0.00 | 15.08±0.00 | 19.44±0.00 | 39.70±0.00 |
| Vsc_mH | 62.00±0.00 | 48.81±0.00 | 56.01±0.00 | 68.53±0.00 | 30.67±0.00 | 25.41±0.00 | 36.92±0.00 | 55.77±0.00 |
| OpVuC | 55.77±0.00 | 36.86±0.00 | 13.63±0.00 | 78.00±0.00 | 17.74±0.00 | 29.69±0.00 | 45.94±0.00 | 47.51±0.00 |
| DCMVC | 31.40±1.27 | 26.22±2.46 | 33.61±2.59 | 67.15±0.42 | 60.71±3.05 | 15.64±0.32 | 61.11±2.62 | 59.83±1.83 |
| DCMVC + Hungarian | 27.63±1.07 | 16.49±1.72 | 22.27±1.69 | 60.41±0.47 | 29.39±0.68 | 12.99±0.31 | 20.68±0.19 | 16.65±0.20 |
| LMTC | 57.39±2.89 | 40.01±4.69 | 33.29±3.81 | 58.75±0.55 | 45.75±0.55 | 23.24±0.45 | 24.01±0.25 | 47.31±0.25 |
| LMTC + Hungarian | 57.79±3.67 | 43.85±2.17 | 37.60±2.84 | 58.98±0.97 | 37.95±0.58 | 23.16±0.58 | 24.59±0.87 | 47.02±2.49 |
| TMSL | 28.68±1.25 | 12.02±1.10 | 23.71±1.08 | 69.33±0.61 | 48.62±0.54 | 21.34±0.36 | OOM | OOM |
| TMSL + Hungarian | 68.40±1.91 | 31.68±0.92 | 25.28±0.81 | 69.46±0.58 | 28.16±0.07 | 19.77±0.27 | OOM | OOM |
| DSTL | 39.59±1.18 | 37.00±0.00 | 23.41±1.99 | 60.53±0.55 | 29.05±0.50 | 15.54±0.33 | 11.64±0.22 | 20.40±0.15 |
| DSTL + Hungarian | 41.17±1.62 | 40.46±0.38 | 24.58±1.13 | 54.13±0.51 | 26.08±0.23 | 14.05±0.50 | 15.32±0.29 | 42.34±0.00 |
| Ours | 69.31±1.32 | 61.20±0.83 | 70.28±0.55 | 85.34±0.42 | 92.09±0.00 | 30.27±0.20 | 74.84±0.94 | 75.50±0.14 |

## 5.2 Compared Methods

Ten state-of-the-art MVC methods are selected as baselines for comparison, including EEOMVC [39], DealMVC [40], MVCAN [41], EBMGC [42], Vsc_mH [43], OpVuC [44], DCMVC [45], LMTC [46], TMSL [47], DSTL [48]. The detailed introductions of them are presented in Appendix A.5.

## 5.3 Experiments Setup

In the experiments, four widely used evaluation metrics are employed to assess the clustering performance of all compared methods: Accuracy (ACC), Normalized Mutual Information (NMI), Adjusted Rand Index (ARI), and F1score. For the proposed method, we conducted a grid search over the set $\{0.001, 0.01, 0.1, 1, 10, 100, 1000, 10000\}$ to determine the best value for each dataset. The unified latent feature dimension $d$ is set to the number of clusters. Regarding the baseline methods, parameters were tuned according to the ranges provided in their respective publicly available source codes, and the best results were selected in the experiments. To mitigate the influence of randomness on the experimental results, each experiment was repeated 20 times, and the mean and variance of the results are reported. All experiments are conducted on a Windows 11 PC equipped with an Intel Core i7-13700F CPU and 64GB RAM.

## 5.4 Results Analysis

To facilitate a fair comparison between the proposed method and existing approaches under the sample non-alignment setting, we fix the sample alignment ratio $\rho$ to 50% in the main experiments.

Table 3: ARI comparison of all methods with and without Hungarian alignment on eight multi-view datasets under a sample alignment ratio $\rho = 50\%$.

| Method | Yale | 3sources | MSRCV | 100leaves | HW | Scene | EMNIST | Hdigit |
|---|---|---|---|---|---|---|---|---|
| EEOMVC | 38.19±0.00 | 29.79±0.00 | 46.02±0.00 | 37.47±0.00 | 86.99±0.00 | 7.76±0.00 | 18.37±0.00 | 53.61±0.00 |
| EEOMVC + Hungarian | 32.46±0.00 | 21.51±0.00 | 45.98±0.00 | 36.53±0.00 | 32.12±0.00 | 6.93±0.00 | 16.33±0.00 | 42.83±0.00 |
| DealMVC | 11.75±0.00 | 2.51±0.44 | 4.49±1.72 | 1.89±0.25 | 21.77±1.06 | 6.57±0.91 | 23.83±1.59 | 38.45±1.65 |
| DealMVC + Hungarian | 3.35±0.00 | 0.24±0.61 | 3.49±0.21 | 2.61±0.49 | 18.77±1.58 | 8.98±2.30 | 15.57±0.49 | 65.76±1.81 |
| MVCAN | 10.54±1.71 | 4.92±1.55 | 35.98±2.09 | 30.37±1.53 | 22.24±0.58 | 16.37±0.60 | 16.49±0.38 | 50.04±3.04 |
| MVCAN + Hungarian | 11.10±1.07 | 29.57±4.45 | 24.14±3.45 | 20.57±1.66 | 30.14±5.39 | 12.52±0.10 | 29.21±12.52 | 42.30±4.52 |
| EBMGC | 15.43±0.00 | 12.29±0.00 | 14.78±0.00 | 15.36±0.00 | 31.06±0.00 | 5.75±0.00 | 12.37±0.00 | 22.04±0.00 |
| EBMGC + Hungarian | 9.89±0.00 | 13.21±0.00 | 17.44±0.00 | 15.36±0.00 | 24.43±0.00 | 8.60±0.00 | 15.06±0.00 | 36.00±0.00 |
| Vsc_mH | 37.27±0.00 | 36.47±0.00 | 45.47±0.00 | 23.39±0.00 | 20.92±0.00 | 13.26±0.00 | 24.93±0.00 | 46.54±0.00 |
| OpVuC | 30.40±0.00 | 33.68±0.00 | 5.49±0.00 | 40.64±0.00 | 10.90±0.00 | 15.39±0.00 | 34.32±0.00 | 38.69±0.00 |
| DCMVC | 4.69±1.05 | 19.77±5.08 | 19.55±2.22 | 29.23±0.91 | 53.01±2.79 | 8.81±0.24 | 48.53±3.72 | 49.42±2.03 |
| DCMVC + Hungarian | 2.01±0.84 | 4.26±1.13 | 14.93±1.54 | 19.50±0.73 | 24.12±0.73 | 6.88±0.24 | 15.51±0.57 | 12.12±0.22 |
| LMTC | 32.32±3.88 | 28.15±4.27 | 24.59±3.88 | 17.24±0.81 | 39.20±0.97 | 11.48±0.43 | 15.42±0.64 | 40.59±0.35 |
| LMTC + Hungarian | 32.71±4.91 | 30.32±2.02 | 27.45±3.00 | 17.58±1.45 | 19.07±1.17 | 11.26±0.43 | 15.09±0.78 | 37.54±2.25 |
| TMSL | 2.64±1.05 | 8.57±1.75 | 15.62±0.83 | 31.41±1.15 | 37.67±0.77 | 10.85±0.31 | OOM | OOM |
| TMSL + Hungarian | 47.94±3.13 | 32.77±1.34 | 16.48±0.77 | 31.38±1.02 | 23.93±0.06 | 9.99±0.21 | OOM | OOM |
| DSTL | 11.47±1.26 | 24.19±0.00 | 12.68±2.02 | 18.49±0.77 | 20.12±0.37 | 6.11±0.20 | 7.25±0.22 | 15.17±0.08 |
| DSTL + Hungarian | 13.70±2.01 | 37.03±0.36 | 14.86±0.97 | 11.85±0.71 | 18.22±0.36 | 5.44±0.25 | 9.31±0.21 | 29.22±0.00 |
| Ours | 48.91±2.65 | 45.98±1.61 | 66.34±0.66 | 59.97±0.95 | 92.49±0.00 | 17.15±0.32 | 65.52±2.12 | 65.53±0.43 |

Table 4: F1score comparison of all methods with and without Hungarian alignment on eight multi-view datasets under a sample alignment ratio $\rho = 50\%$.

| Method | Yale | 3sources | MSRCV | 100leaves | HW | Scene | EMNIST | Hdigit |
|---|---|---|---|---|---|---|---|---|
| EEOMVC | 42.10±0.00 | 44.77±0.00 | 53.93±0.00 | 38.17±0.00 | 88.29±0.00 | 15.02±0.00 | 28.53±0.00 | 58.97±0.00 |
| EEOMVC + Hungarian | 36.75±0.00 | 36.99±0.00 | 53.83±0.00 | 37.28±0.00 | 41.02±0.00 | 14.41±0.00 | 26.56±0.00 | 49.21±0.00 |
| DealMVC | 25.78±0.00 | 31.29±1.10 | 26.27±2.57 | 6.68±0.15 | 32.01±0.87 | 14.88±0.95 | 32.84±1.17 | 44.73±1.50 |
| DealMVC + Hungarian | 23.89±0.00 | 30.27±2.35 | 25.19±1.20 | 7.66±1.06 | 30.95±1.94 | 18.86±1.72 | 27.34±0.04 | 69.41±1.61 |
| MVCAN | 25.09±1.61 | 29.14±1.02 | 47.00±1.67 | 37.49±1.29 | 34.95±0.45 | 22.78±0.41 | 25.18±0.26 | 57.33±1.99 |
| MVCAN + Hungarian | 25.35±1.43 | 54.48±2.79 | 37.97±2.34 | 28.31±1.85 | 39.76±3.43 | 19.98±0.15 | 37.12±11.11 | 49.72±3.79 |
| EBMGC | 20.59±0.00 | 29.02±0.00 | 26.60±0.00 | 16.16±0.00 | 37.92±0.00 | 12.13±0.00 | 21.13±0.00 | 29.84±0.00 |
| EBMGC + Hungarian | 15.38±0.00 | 29.77±0.00 | 28.90±0.00 | 16.16±0.00 | 31.95±0.00 | 14.80±0.00 | 23.55±0.00 | 42.40±0.00 |
| Vsc_mH | 41.70±0.00 | 52.20±0.00 | 53.91±0.00 | 24.41±0.00 | 29.42±0.00 | 20.55±0.00 | 32.83±0.00 | 51.99±0.00 |
| OpVuC | 34.92±0.00 | 46.62±0.00 | 20.44±0.00 | 41.36±0.00 | 20.65±0.00 | 21.59±0.00 | 41.09±0.00 | 44.99±0.00 |
| DCMVC | 19.48±0.98 | 41.10±2.31 | 35.18±1.69 | 35.08±0.61 | 59.95±3.01 | 15.57±0.23 | 55.15±3.28 | 56.02±2.45 |
| DCMVC + Hungarian | 17.08±0.81 | 36.92±1.97 | 30.19±1.32 | 25.58±0.63 | 33.17±0.62 | 13.94±0.19 | 25.51±0.33 | 21.22±0.23 |
| LMTC | 36.59±3.61 | 42.40±3.28 | 35.28±3.33 | 18.07±0.81 | 45.29±0.87 | 17.65±0.41 | 24.48±0.42 | 46.62±0.31 |
| LMTC + Hungarian | 36.95±4.57 | 43.81±1.57 | 37.70±2.54 | 18.41±1.43 | 29.32±0.89 | 17.43±0.40 | 24.34±0.42 | 43.94±2.04 |
| TMSL | 9.04±0.97 | 32.77±2.01 | 27.43±0.74 | 32.09±1.14 | 44.06±0.68 | 17.46±0.34 | OOM | OOM |
| TMSL + Hungarian | 51.19±2.91 | 47.76±1.06 | 28.15±0.67 | 32.07±1.01 | 31.54±0.06 | 16.37±0.20 | OOM | OOM |
| DSTL | 17.50±1.09 | 47.67±0.00 | 25.27±1.75 | 19.35±0.75 | 28.38±0.35 | 12.65±0.20 | 16.81±0.16 | 23.70±0.07 |
| DSTL + Hungarian | 19.25±1.87 | 51.00±0.28 | 26.87±0.84 | 12.79±0.71 | 26.72±0.34 | 12.06±0.24 | 18.65±0.22 | 37.49±0.00 |
| Ours | 52.17±2.43 | 56.59±1.34 | 71.03±0.57 | 60.37±0.94 | 93.24±0.00 | 22.94±0.29 | 69.12±1.85 | 69.16±0.37 |

Due to space constraints, results under other alignment ratios are provided in the Appendix and can be found in Tables 7-8 for reference. Notably, some baselines are not directly applicable to the non-aligned scenario. For fair evaluation, we apply the Hungarian algorithm to align the data before using these methods. Clustering results under the four evaluation metrics are shown in Tables 1-4 with the best and the second results highlighted in **bold** and underlined respectively. Methods that encounter memory overflow are marked as OOM. Based on the results reported in the Tables, several key observations can be obtained:

(1) The proposed algorithm consistently outperforms most baseline methods, including those using Hungarian-based sample alignment. For example, on the EMNIST and MSRCV datasets, it achieves ACC improvements of 10.66% and 17.83% over the second-best methods, EEOMVC and DCMVC, respectively. Similar gains are observed across other datasets, highlighting the method's effectiveness in capturing true cross-view sample correspondences and enhancing clustering performance.

(2) Our method is superior to existing methods such as Vsc_mH and OpVuC, which are designed for non-aligned sample clustering. These two kind of methods rely on mining alignment relationships directly from raw features without explicitly modeling the structural hierarchy within each view. Moreover, they employ a hard matching strategy, determining class correspondences based on pairwise sample similarity. Due to high intra-class similarity, this often results in unstable alignment matrices, adversely affecting algorithm convergence and leading to performance fluctuations.

(3) Compared with deep clustering methods such as DealMVC, MVCAN, and DCMVC, the proposed method demonstrates notable advantages. Although deep neural networks possess strong representation capabilities, they often rely on the assumption of consistent semantic information

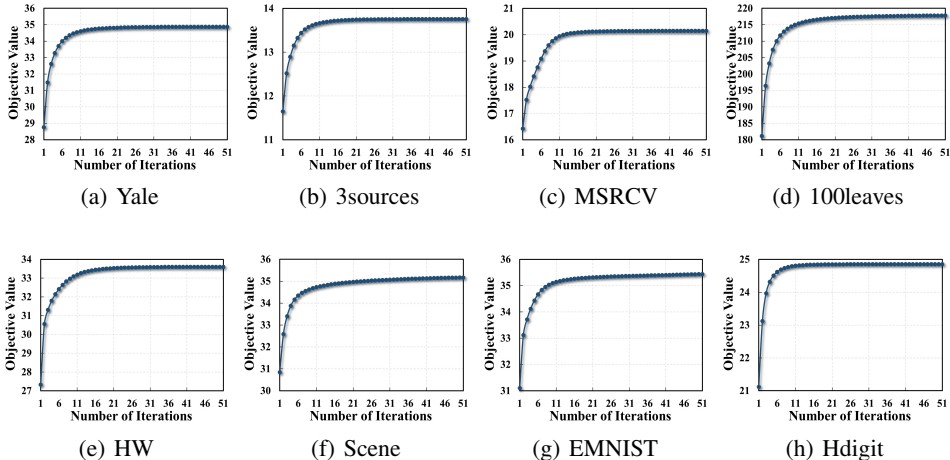

Figure 2: The objective function values of the proposed method across iterations.

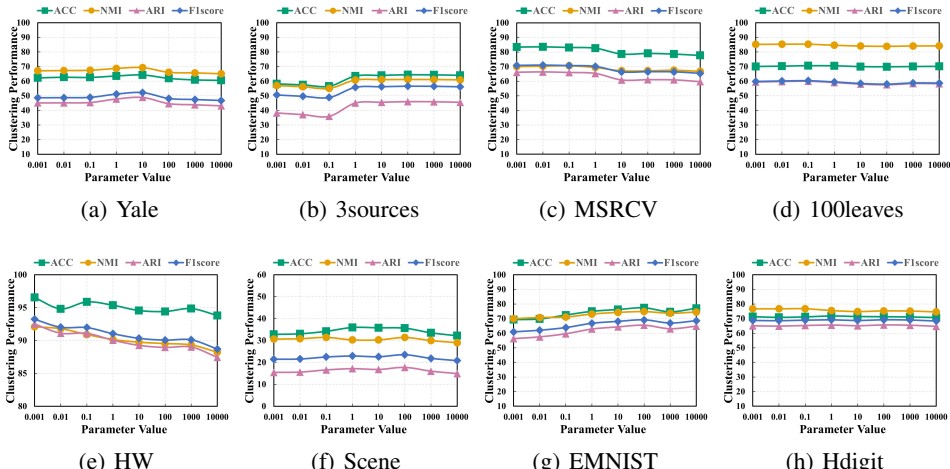

Figure 3: Clustering performance of the proposed method with varying values of the parameter $\lambda$.

across views. This assumption breaks down in the presence of sample misalignment, resulting in inconsistent feature learning and diminished clustering performance. Moreover, the use of the Hungarian algorithm for late fusion alignment does not consistently lead to performance gains and can even degrade results. This may be due to incorrect alignments introducing noisy or misleading information, ultimately impairing the effectiveness of the model.

### 5.5 Convergence and Parameter Sensitivity Analysis

In the previous section, we theoretically established that the proposed algorithm converges within a finite number of iterations. In this section, we further verify the convergence behavior empirically. The corresponding experimental results are illustrated in Fig. 2. As shown in the figure, the proposed method typically converges within approximately 10 iterations across all datasets, which empirically confirms its favorable convergence properties.

The results of our proposed method across varying $\lambda$ values are presented in Fig. 3. Overall, the method demonstrates strong robustness to $\lambda$, with stable performance on most datasets. Notably, fluctuations on datasets like 3sources suggest higher sensitivity, may be can attributed to significant semantic divergence among views, which underscores the importance of appropriately weighting structural similarity.

## 5.6 Ablation Studies

We conduct ablation studies to assess the contribution of the proposed cross-view structural similarity module to clustering performance. Specifically, we denote the model without this module as SSA-MVC w/o CVS. The results, shown in Fig. 4, indicate that incorporating the module consistently improves sample alignment and clustering performance across most datasets. These findings highlight the effectiveness of the module and its integral role in the overall framework.

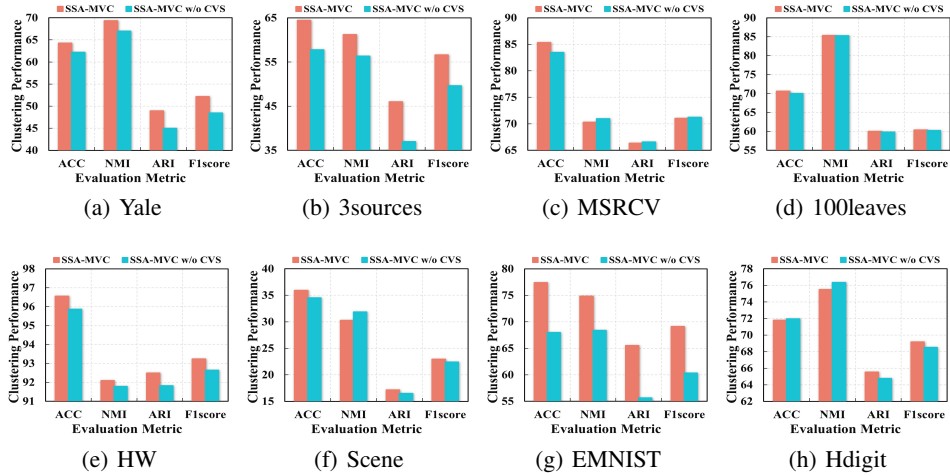

Figure 4: Clustering performance of the proposed method and its variant on eight multi-view datasets.

## 5.7 Effectiveness of the Alignment Strategy

To evaluate the scalability of our proposed method, we assess its effectiveness on the clustering algorithms that do not inherently handle sample misalignment. Specifically, under an alignment ratio of $\rho = 50\%$, we use $\mathbf{M}$ to realign the originally non-aligned 100leaves multi-view data and compare the performance of several baseline algorithms on both the original and the realigned 100leaves. As shown in Table 5, our method can benefit the clustering performance of these algorithms in the non-aligned setting, demonstrating its effectiveness and potential for generalization to other methods.

Table 5: Results of competitors on the 100leaves under a sample alignment ratio of $\rho = 50\%$.

| Setting | Metric | DealMVC | MVCAN | EBMGC | DCMVC | LMTC | TMSL | DSTL |
|---------|--------|---------|-------|-------|-------|------|------|------|
| ACC | Unaligned | 7.69±0.00 | 49.51±1.24 | 33.94±0.00 | 48.83±0.83 | 35.58±0.94 | 47.47±1.40 | 36.87±1.42 |
| | Aligned+Ours | 12.42±0.52 | 48.81±1.28 | 43.06±0.00 | 53.75±0.89 | 40.82±1.31 | 48.18±1.37 | 35.60±0.90 |
| NMI | Unaligned | 25.34±0.44 | 69.50±0.95 | 58.22±0.00 | 67.15±0.42 | 58.75±0.55 | 69.33±0.61 | 60.53±0.55 |
| | Aligned+Ours | 38.01±0.46 | 69.95±0.59 | 64.52±0.00 | 72.10±0.49 | 64.58±0.80 | 70.54±0.57 | 61.09±0.53 |
| ARI | Unaligned | 1.89±0.25 | 30.37±1.53 | 15.36±0.00 | 29.23±0.91 | 17.24±0.81 | 31.41±1.15 | 18.49±0.77 |
| | Aligned+Ours | 5.16±0.12 | 30.58±0.99 | 24.62±0.00 | 36.50±0.87 | 24.46±1.24 | 32.67±1.17 | 19.38±0.68 |
| F1score | Unaligned | 6.68±0.15 | 37.49±1.29 | 16.16±0.00 | 35.08±0.61 | 18.07±0.81 | 32.09±1.14 | 19.35±0.75 |
| | Aligned+Ours | 11.47±0.14 | 37.74±0.93 | 25.33±0.00 | 41.38±0.85 | 25.21±1.23 | 33.34±1.15 | 20.23±0.67 |

## 6 Conclusion

This paper proposes a scalable multi-view clustering algorithm to tackle sample non-alignment. By selecting a baseline view via the CLM algorithm and leveraging structural similarities between aligned and non-aligned samples, the method guides cross-view alignment and integrates the resulting alignment matrix into a late fusion clustering framework. Experiments on eight benchmark datasets validate the effectiveness of the proposed method.

## Acknowledgments

This work was supported by the National Natural Science Foundation of China (NO. 62325604, 62276271, 62441618).

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

# A  Appendix

## A.1  Limitations

As revealed by the construction process of the view-specific structural representations, the proposed method exhibits certain limitations when applied to scenarios where samples across views are completely unaligned. In such cases, a feasible approach is to explore the cross-view structural correlations by computing similarities between all samples in the to-be-aligned view and those in a designated baseline view. This cross-view relational information can then be integrated into the overall clustering framework after the main convergence process. Accordingly, future research will focus on designing multi-view clustering algorithms that are specifically tailored to handle fully unaligned sample scenarios.

## A.2  The Pseudo Code of the Proposed Method

The detailed algorithm optimization processes are presented in the following.

---

**Algorithm 1** The Algorithm of SSA-MVC.

---

1: **Input**: Unaligned multi-view data $\{\mathbf{X}^v\}_{v=1}^V$, the number of clusters $k$, the unified feature dimension $d$, and the hyper-parameter $\lambda$.
2: Construct the cross-view similarity graph $\{\mathbf{S}^v\}_{v=1}^V$ via Eqs. (3-(6)).
3: Initialize $\{\mathbf{R}^v\}_{v=1}^V$, $\{\mathbf{M}^v\}_{v=1}^V$, $\{\alpha_v\}_{v=1}^V$.
4: **while** not converge **do**
5:     Update $\mathbf{F}^*$ via Eq. (8).
6:     Update $\{\mathbf{R}^v\}_{v=1}^V$ via Eq. (10).
7:     Update $\{\alpha_v\}_{v=1}^V$ via Eq. (11).
8:     Update $\{\mathbf{M}^v\}_{v=1}^V$ via Eq. (13).
9: **end while**
10: Conduct $k$-means clustering algorithm on the consensus partition $\mathbf{F}^*$.
11: **Output**: Clustering results $\mathbf{Y}$.

---

## A.3  Proof of Theorem 1

*Proof.* The proof can be divided into two parts, i.e., the objective function is upper bounded, and it is monotonically increasing.

1) The objective function is upper bounded.

Given that $\mathbf{M}^t = \mathbf{I}$, the overall objective function in Eq. (7) can be simplified as follows:

$$
\max_{\mathbf{R}^v, \mathbf{F}^*, \mathbf{M}^v, \alpha_v} \sum_{v=1}^V \mathrm{Tr}\big(\mathbf{F}^{*\top} \alpha_v \mathbf{C}^v \mathbf{F}^v \mathbf{R}^v\big) + \lambda \sum_{v=1}^V \mathrm{Tr}\big(\mathbf{M}^{v\top} \mathbf{S}^v\big)
$$
$$
s.t.\ \mathbf{C}^v = \begin{bmatrix} \mathbf{I} & \mathbf{0} \\ \mathbf{0} & \mathbf{M}^v \end{bmatrix}, \mathbf{F}^{*\top}\mathbf{F}^* = \mathbf{I}, \mathbf{R}^{v\top}\mathbf{R}^v = \mathbf{I}, \sum_{v=1}^V \alpha_v^2 = 1, \mathbf{M}^{v\top}\mathbf{M}^v = \mathbf{I}. \tag{14}
$$

For any two distinct views $v$ and $v'$, where $v \neq v'$, the following inequality holds:

$$
\begin{aligned}
&\mathrm{Tr}\left((\alpha_v \mathbf{C}^v \mathbf{F}^v \mathbf{R}^v)^\top (\alpha_{v'} \mathbf{C}^{v'} \mathbf{F}^{v'} \mathbf{R}^{v'})\right) \\
&\leq \mathrm{Tr}\left((\mathbf{C}^v \mathbf{F}^v \mathbf{R}^v)^\top (\mathbf{C}^{v'} \mathbf{F}^{v'} \mathbf{R}^{v'})\right) \\
&\leq \frac{1}{2}\left(\mathrm{Tr}\left((\mathbf{C}^v \mathbf{F}^v \mathbf{R}^v)^\top (\mathbf{C}^v \mathbf{F}^v \mathbf{R}^v)\right) + \mathrm{Tr}\left((\mathbf{C}^{v'} \mathbf{F}^{v'} \mathbf{R}^{v'})^\top (\mathbf{C}^{v'} \mathbf{F}^{v'} \mathbf{R}^{v'})\right)\right) \\
&= d
\end{aligned} \tag{15}
$$

Based on the above inequality, we can further derive that:

$$\mathrm{Tr}\left(\mathbf{F}^{*\top}\sum_{v=1}^{V}\alpha_v\mathbf{C}^v\mathbf{F}^v\mathbf{R}^v\right)$$

$$\leq \frac{1}{2}\left(\mathrm{Tr}(\mathbf{F}^{*\top}\mathbf{F}^*) + \mathrm{Tr}\left(\left(\sum_{v=1}^{V}\alpha_v\mathbf{C}^v\mathbf{F}^v\mathbf{R}^v\right)^{\top}\left(\sum_{v'=1}^{V}\alpha_{v'}\mathbf{C}^{v'}\mathbf{F}^{v'}\mathbf{R}^{v'}\right)\right)\right) \qquad (16)$$

$$\leq \frac{1}{2}(d + dn^2)$$

where $d$ is the unified feature dimension, and $n$ is the number of samples. Moreover, for the regularization term, we have:

$$\lambda\,\mathrm{Tr}(\mathbf{M}^{v\top}\mathbf{S}^v) \leq \frac{\lambda}{2}\left(\mathrm{Tr}(\mathbf{M}^{v\top}\mathbf{M}^v) + \mathrm{Tr}(\mathbf{S}^{v\top}\mathbf{S}^v)\right) = \frac{\lambda}{2}\left(n_2 + \mathrm{Tr}(\mathbf{S}^{v\top}\mathbf{S}^v)\right) \qquad (17)$$

where $n_2$ is a number of unaligned samples. Since $\mathrm{Tr}(\mathbf{S}^{v\top}\mathbf{S}^v)$ is a constant and $\lambda$ is fixed, the term $\lambda\,\mathrm{Tr}(\mathbf{M}^{v\top}\mathbf{S}^v)$ is upper bounded.

Therefore, the overall objective function in Eq. (7) is guaranteed to be upper bounded.

2) The objective function is monotonically increasing.

In the aforementioned optimization procedure, it is apparent that the sub-problems involving the variables $\mathbf{F}^*$, $\mathbf{R}^v$, and $\mathbf{M}^v$ respectively, reduce to classical Orthogonal Procrustes Problems. Throughout the iterative solution process, the objective function values associated with these variables exhibit a monotonically non-decreasing trend. Additionally, the optimization with respect to the variable $\alpha$ constitutes a standard linear objective maximization problem under quadratic constraints, which similarly guarantees a monotonically non-decreasing progression of the objective value during updates. Let $\Theta(\mathbf{F}^*, \mathbf{M}^v, \mathbf{R}^v, \alpha_v)$ represent a simplified form of the objective function defined in Eq. (7). Thus, the following inequalities hold:

$$\Theta\big(\{\mathbf{F}^*\}^{(p)}, \{\mathbf{M}^v\}^{(p)}, \{\mathbf{R}^v\}^{(p)}, \{\alpha_v\}^{(p)}\big) \leq \Theta\big(\{\mathbf{F}^*\}^{(p+1)}, \{\mathbf{M}^v\}^{(p+1)}, \{\mathbf{R}^v\}^{(p+1)}, \{\alpha_v\}^{(p+1)}\big). \tag{18}$$

where the superscript $(p)$ and $(p+1)$ denote the number of iterations.

Therefore, based on the aforementioned properties, we can conclude that the proposed algorithm is guaranteed to converge during the optimization process. $\qquad\square$

### A.4 Datasets Description

In our experiments, eight multi-view benchmark datasets are used to verify the effectiveness of our proposed method, including Yale[2], 3sources[3], MSRCV[4], 100leaves[5], HW[6], Scene [49], EMNIST[7], and Hdigit[8]. In the following, we will give a detailed introduction to them.

**Yale:** This dataset comprises 165 samples distributed across 15 distinct classes. Each sample is characterized by three heterogeneous feature sets: a 4096-dimensional Intensity descriptor, a 3304-dimensional Local Binary Pattern (LBP) descriptor, and a 6750-dimensional Gabor descriptor.

**3sources:** It comprises 169 samples collected from three distinct news media sources: BBC, Reuters, and The Guardian. Each sample is represented by three views, corresponding to the textual content extracted from each respective source. The feature dimensions for these views are 3560, 3631, and 3068 for BBC, Reuters, and The Guardian, respectively. All samples are annotated with one of six semantic classes.

---

[2] https://vision.ucsd.edu/content/yale-face-database
[3] http://mlg.ucd.ie/datasets/3sources.html
[4] https://mldta.com/dataset/msrc-v1/
[5] https://archive.ics.uci.edu/ml/datasets/Onehundred+plant+species+leaves+data+set
[6] https://archive.ics.uci.edu/ml/datasets/Multiple+Features
[7] https://www.nist.gov/itl/products-and-services/emnist-dataset
[8] https://cs.nyu.edu/ roweis/data.html

Table 6: Summary of eight benchmark multi-view datasets.

| Datasets | #Classes | #Samples | #Views | #Feature Dimensionalities |
|----------|----------|----------|--------|---------------------------|
| Yale | 15 | 165 | 3 | 4096; 3304; 6750 |
| 3sources | 6 | 169 | 3 | 3560; 3631; 3068 |
| MSRCV | 7 | 210 | 5 | 24; 576; 512; 256; 254 |
| 100leaves | 100 | 1600 | 3 | 64; 64; 64; |
| HW | 10 | 2000 | 6 | 216; 76; 64; 6; 240; 47 |
| Scene | 15 | 4485 | 3 | 20; 59; 40 |
| EMNIST | 10 | 10000 | 4 | 576; 944; 512; 640 |
| Hdigit | 10 | 10000 | 2 | 784; 256 |

**MSRCV:** It comprises 210 image samples, each labeled with one of seven semantic classes. For each sample, a five-view feature representation is provided to capture diverse visual characteristics. Specifically, the dataset includes the following feature descriptors: 24-dimensional Color Moments (CM), 576-dimensional Histogram of Oriented Gradients (HOG), 512-dimensional GIST, 256-dimensional Local Binary Patterns (LBP), and 254-dimensional Gabor Energy-based Texture (GENT) features.

**100leaves:** The dataset comprises 1600 samples distributed across 100 distinct leaf species. Each sample is characterized by three complementary feature views: a 64-dimensional Texture Histogram (TH), a 64-dimensional Fourier Shape-based Metric (FSM), and a 64-dimensional Statistical Descriptor (SD). These multi-view features encapsulate diverse morphological and structural characteristics of the leaves, rendering the dataset highly suitable for the evaluation of multi-view learning and clustering algorithms.

**HW:** The dataset comprises 2000 samples, each annotated with one of ten distinct class labels. It encompasses six heterogeneous views, each representing diverse feature modalities extracted from the same set of samples. Specifically, these views include: 76-dimensional FOU features, 216-dimensional FAC features, 64-dimensional KAR features, 240-dimensional PIX features, 47-dimensional ZER features, and 6-dimensional MOR features.

**Scene:** The dataset comprises 4485 samples distributed across 15 distinct scene categories. Each sample is described by three complementary visual modalities: a 1800-dimensional GIST descriptor capturing the global spatial layout, a 1180-dimensional PHOG feature encoding local shape information, and a 1240-dimensional LBP representation characterizing texture patterns.

**EMNIST:** The dataset comprises 10000 samples distributed across 10 distinct classes. Each sample is characterized by four heterogeneous views, each providing complementary information derived from different feature sets. Specifically, the dimensionalities of the features corresponding to the four views are 576, 944, 512, and 640, respectively.

**Hdigit:** The dataset comprises 5000 handwritten digit images, representing the ten classes from 0 to 9. These samples are drawn from two distinct sources: the MNIST and USPS digit datasets. By integrating variations in handwriting styles and image resolutions inherent to both domains, the dataset offers a comprehensive and challenging benchmark for evaluating digit recognition algorithms.

### A.5 Compared Methods Introduction

In this section, the specific introduction of ten state-of-the-art multi-view clustering methods is illustrated in the following.

**EEOMVC** (TNNLS 23) [39]: This method efficiently performs one-step multi-view clustering by constructing anchor-based similarity graphs to learn unified latent partition representations, enabling direct extraction of discrete clustering labels. By integrating latent information fusion and clustering into a joint framework, it significantly reduces computational complexity while improving clustering accuracy on large-scale datasets.

**DealMVC**(ACM MM 23) [40]: The proposed method addresses the limitation of existing multi-view clustering models by aligning similar yet distinct samples across views through dual contrastive

Table 7: Clustering performance of all compared methods on eight multi-view datasets under a sample alignment ratio of $\rho = 25\%$.

| Dataset | Metric | EEOMVC | DealMVC | MVCAN | EBMGC | Vsc_mH | OpVuC | DCMVC | LMTC | TMSL | DSTL | Ours |
|---|---|---|---|---|---|---|---|---|---|---|---|---|
| Yale | ACC | 47.88±0.00 | 10.30±0.00 | 26.18±1.94 | 27.27±0.00 | 43.64±0.00 | 24.24±0.00 | 22.91±1.04 | 47.55±3.35 | 22.21±1.13 | 30.27±1.45 | 63.70±2.15 |
| | NMI | 52.06±0.00 | 8.90±0.00 | 33.85±2.67 | 33.22±0.00 | 48.62±0.00 | 27.76±0.00 | 27.42±1.11 | 53.72±2.81 | 26.70±1.08 | 35.32±1.33 | 66.45±1.30 |
| | ARI | 27.86±0.00 | 0.00±0.00 | 5.96±1.82 | 5.51±0.00 | 20.60±0.00 | 2.12±0.00 | 1.71±0.70 | 27.03±3.46 | 0.93±0.68 | 7.68±1.12 | 45.33±2.21 |
| | F1score | 32.37±0.00 | 15.60±0.00 | 20.39±1.50 | 11.27±0.00 | 25.61±0.00 | 8.83±0.00 | 16.61±0.60 | 31.64±3.24 | 7.18±0.63 | 13.72±1.06 | 48.75±2.06 |
| 3sources | ACC | 53.25±0.00 | 31.48±0.95 | 28.69±1.26 | 33.14±0.00 | 53.85±0.00 | 44.38±0.00 | 33.73±2.97 | 45.62±3.68 | 36.42±1.74 | 43.28±0.29 | 62.31±0.79 |
| | NMI | 29.93±0.00 | 4.76±0.17 | 9.24±2.28 | 13.84±0.00 | 33.23±0.00 | 25.19±0.00 | 15.43±2.18 | 33.76±5.62 | 5.84±0.73 | 20.33±0.56 | 55.65±0.18 |
| | ARI | 26.50±0.00 | 0.04±0.53 | 1.20±1.28 | 4.96±0.00 | 22.59±0.00 | 25.16±0.00 | 5.30±3.33 | 23.78±5.21 | 1.04±0.94 | 14.00±0.82 | 41.89±0.69 |
| | F1score | 41.65±0.00 | 31.68±0.90 | 27.21±1.47 | 23.09±0.00 | 46.29±0.00 | 40.12±0.00 | 34.09±3.03 | 38.77±4.29 | 30.37±1.17 | 40.04±0.16 | 53.24±0.58 |
| MSRCV | ACC | 60.95±0.00 | 33.62±2.48 | 49.62±6.05 | 26.67±0.00 | 55.24±0.00 | 26.67±0.00 | 39.95±2.33 | 47.40±2.12 | 32.29±0.63 | 35.88±3.02 | 80.12±0.43 |
| | NMI | 47.27±0.00 | 13.07±1.64 | 39.58±6.76 | 10.62±0.00 | 44.69±0.00 | 7.59±0.00 | 30.36±1.89 | 33.35±2.18 | 11.74±0.72 | 18.68±1.59 | 66.45±0.94 |
| | ARI | 37.21±0.00 | 6.00±2.03 | 25.61±5.75 | 3.37±0.00 | 33.00±0.00 | 3.21±0.00 | 15.15±1.82 | 21.43±1.78 | 5.04±0.33 | 9.57±1.38 | 61.49±0.83 |
| | F1score | 46.16±0.00 | 23.55±1.19 | 40.02±5.38 | 16.78±0.00 | 42.50±0.00 | 18.14±0.00 | 31.63±1.36 | 32.47±1.46 | 18.52±0.28 | 22.32±1.23 | 66.87±0.71 |
| 100leaves | ACC | 39.69±0.00 | 4.14±0.15 | 29.93±0.89 | 21.44±0.00 | 24.69±0.00 | 18.50±0.00 | 31.82±0.60 | 22.92±0.96 | 43.50±0.92 | 22.23±0.76 | 66.51±0.92 |
| | NMI | 59.31±0.00 | 13.66±0.12 | 58.28±0.65 | 50.90±0.00 | 54.50±0.00 | 46.59±0.00 | 57.20±0.59 | 50.21±0.65 | 68.98±0.45 | 50.93±0.50 | 83.11±0.27 |
| | ARI | 19.07±0.00 | 0.19±0.04 | 13.66±0.70 | 6.24±0.00 | 11.50±0.00 | 6.67±0.00 | 15.35±0.79 | 7.93±0.85 | 29.30±0.97 | 7.38±0.43 | 55.29±0.91 |
| | F1score | 19.92±0.00 | 3.87±0.05 | 21.15±0.67 | 7.12±0.00 | 12.63±0.00 | 7.93±0.00 | 21.85±0.54 | 8.86±0.84 | 30.00±0.96 | 8.37±0.42 | 55.74±0.90 |
| HW | ACC | 92.05±0.00 | 27.58±0.84 | 28.55±0.03 | 20.80±0.00 | 19.25±0.00 | 19.20±0.00 | 75.25±1.67 | 45.59±2.36 | 60.67±0.48 | 31.56±0.66 | 96.35±0.00 |
| | NMI | 83.78±0.00 | 7.85±0.28 | 8.70±0.05 | 6.75±0.00 | 10.27±0.00 | 4.44±0.00 | 73.09±0.88 | 29.00±1.70 | 47.52±0.26 | 16.38±0.39 | 91.59±0.00 |
| | ARI | 83.22±0.00 | 4.96±0.54 | 5.06±0.03 | 3.37±0.00 | 1.55±0.00 | 1.85±0.00 | 65.60±1.36 | 21.06±1.75 | 37.85±0.40 | 9.85±0.30 | 92.08±0.00 |
| | F1score | 84.89±0.00 | 16.40±0.10 | 15.58±0.02 | 12.99±0.00 | 17.32±0.00 | 12.53±0.00 | 70.78±1.28 | 29.00±1.57 | 44.07±0.35 | 18.92±0.28 | 92.87±0.00 |
| Scene | ACC | 23.90±0.00 | 15.04±0.37 | 26.77±0.37 | 14.25±0.00 | 29.68±0.00 | 15.92±0.00 | 17.08±0.19 | 26.99±0.55 | 25.56±0.70 | 16.60±0.47 | 32.95±0.78 |
| | NMI | 16.45±0.00 | 3.48±0.13 | 30.82±0.62 | 3.76±0.00 | 28.68±0.00 | 6.26±0.00 | 5.64±0.26 | 22.17±0.39 | 18.65±0.49 | 10.07±0.22 | 27.79±0.32 |
| | ARI | 7.24±0.00 | 1.45±0.06 | 14.49±0.33 | 1.44±0.00 | 15.06±0.00 | 2.25±0.00 | 2.47±0.14 | 10.65±0.36 | 9.03±0.52 | 3.66±0.15 | 14.36±0.32 |
| | F1score | 14.41±0.00 | 10.70±0.21 | 21.28±0.27 | 8.12±0.00 | 21.56±0.00 | 10.48±0.00 | 9.72±0.14 | 16.87±0.34 | 15.81±0.54 | 10.52±0.16 | 20.35±0.29 |
| EMNIST | ACC | 36.53±0.00 | 43.29±1.86 | 26.72±0.48 | 17.46±0.00 | 46.69±0.00 | 47.66±0.00 | 59.61±3.04 | 30.30±0.43 | OOM | 19.69±0.18 | 75.43±4.25 |
| | NMI | 17.58±0.00 | 30.64±3.19 | 5.79±0.10 | 3.64±0.00 | 39.73±0.00 | 41.04±0.00 | 60.23±3.14 | 10.11±0.27 | OOM | 4.33±0.07 | 70.66±1.87 |
| | ARI | 11.10±0.00 | 22.02±2.45 | 3.91±0.12 | 1.90±0.00 | 27.02±0.00 | 28.33±0.00 | 48.39±3.30 | 6.99±0.20 | OOM | 2.28±0.05 | 61.67±3.56 |
| | F1score | 21.17±0.00 | 32.39±1.43 | 13.74±0.09 | 11.70±0.00 | 34.59±0.00 | 35.65±0.00 | 55.20±3.29 | 16.32±0.18 | OOM | 12.30±0.07 | 65.65±3.13 |
| Hdigit | ACC | 64.76±0.00 | 41.34±1.18 | 60.24±6.04 | 20.82±0.00 | 58.38±0.00 | 24.09±0.00 | 57.54±2.22 | 53.64±1.48 | OOM | 30.10±0.06 | 71.65±4.61 |
| | NMI | 71.68±0.00 | 15.96±1.27 | 55.53±3.46 | 5.09±0.00 | 48.48±0.00 | 6.22±0.00 | 55.37±0.97 | 45.56±2.18 | OOM | 11.77±0.05 | 74.87±1.00 |
| | ARI | 53.06±0.00 | 13.21±1.27 | 43.99±5.45 | 2.93±0.00 | 39.14±0.00 | 3.89±0.00 | 42.73±0.81 | 34.89±1.42 | OOM | 8.02±0.03 | 63.35±3.00 |
| | F1score | 58.55±0.00 | 21.94±1.09 | 51.04±4.18 | 12.63±0.00 | 45.36±0.00 | 13.71±0.00 | 50.69±1.05 | 41.56±1.29 | OOM | 17.26±0.03 | 67.28±2.59 |

calibration losses at both global and local levels. This approach effectively integrates cross-view feature similarity and reliable class information, enhancing clustering performance and robustness.

**MVCAN** (CVPR 24) [41]: MVCAN is a theoretically grounded deep multi-view clustering method designed to mitigate the impact of noisy views by allowing unshared parameters and inconsistent clustering predictions across views. It employs a two-level iterative optimization to enhance representation learning, achieving multi-view consistency, complementarity, and robustness to noise.

**EBMGC** (TPAMI 24) [42]: This method effectively leverages consistent neighbor information across multiple views through a novel Cross-view Good Neighbors Voting module, while a balanced regularization term based on the p-power function adapts clustering to diverse data distributions. By incorporating graph coarsening and an accelerated coordinate descent algorithm, this method achieves superior clustering performance with high efficiency.

**Vsc_mH** (Neural Networks 24) [43]: This method effectively addresses the View-shuffled Problem by simultaneously establishing cross-view correspondences through a global alignment and modified Hungarian algorithm, and performing clustering via matrix factorization. This integrated approach enables robust clustering on shuffled multi-view data with varying alignment ratios, supported by both theoretical convergence guarantees and strong empirical performance.

**OpVuC** (TMM 24) [44]: This method simultaneously addresses instance alignment and clustering within a unified framework, effectively handling fully unaligned multi-view data without relying on any pre-aligned samples. By leveraging a novel global-local alignment strategy grounded in geometric invariance and a relaxed k-means clustering approach, OpVuC robustly processes data at any alignment level, demonstrating superior performance across benchmark datasets.

**DCMVC** (TIP 24) [45]: This paper introduces a deep multi-view clustering network with a dual contrastive mechanism that simultaneously enhances inter-cluster separation and within-cluster compactness to learn clustering-friendly representations. By integrating dynamic cluster diffusion and neighbor-guided positive alignment losses, it effectively fuses multi-view features into discriminative consensus representations, achieving superior clustering performance.

**LMTC** (CVPR 25) [46]: This method removes the tensor rotation trick to avoid inadvertent label information and introduces a large-scale multi-view tensor clustering approach that incorporates pair-wise similarities via an implicit linear kernel. This results in an efficient, linear-complexity algorithm that effectively improves clustering performance without relying on sequential data order.

**TMSL** (KBS 25) [47]: This method enhances traditional tensor-based multi-view clustering by leveraging tensor low-rank representation to capture the intrinsic data structure, resulting in a more

Table 8: Clustering performance of all compared methods on eight multi-view datasets under a sample alignment ratio of $\rho = 75\%$.

| Dataset | Metric | EEOMVC | DealMVC | MVCAN | EBMGC | Vsc_mH | OpVuC | DCMVC | LMTC | TMSL | DSTL | Ours |
|---|---|---|---|---|---|---|---|---|---|---|---|---|
| Yale | ACC | 58.79±0.00 | 35.76±0.00 | 47.27±0.77 | 47.88±0.00 | 56.36±0.00 | 47.27±0.00 | 41.03±1.72 | 60.03±4.15 | 33.97±2.32 | 46.30±2.62 | 63.91±2.99 |
| | NMI | 61.44±0.00 | 41.67±0.00 | 51.80±1.08 | 51.58±0.00 | 58.05±0.00 | 49.86±0.00 | 45.07±1.46 | 62.21±3.24 | 39.04±1.76 | 49.02±1.63 | 67.88±1.77 |
| | ARI | 38.57±0.00 | 17.33±0.00 | 24.80±1.31 | 24.29±0.00 | 32.25±0.00 | 23.35±0.00 | 18.16±1.75 | 40.19±4.51 | 11.37±1.75 | 22.57±2.26 | 47.71±2.28 |
| | F1score | 42.49±0.00 | 30.62±0.00 | 37.90±0.62 | 28.92±0.00 | 36.94±0.00 | 28.43±0.00 | 32.71±1.65 | 43.94±4.21 | 17.35±1.67 | 27.71±2.03 | 51.02±2.12 |
| 3sources | ACC | 65.09±0.00 | 30.30±0.24 | 43.59±2.54 | 42.60±0.00 | 63.31±0.00 | 53.25±0.00 | 63.31±5.91 | 50.33±1.44 | 52.60±2.17 | 54.44±0.00 | 65.24±0.26 |
| | NMI | 49.55±0.00 | 5.23±0.92 | 28.13±3.29 | 28.48±0.00 | 55.46±0.00 | 38.55±0.00 | 44.27±4.34 | 46.97±2.71 | 30.84±1.01 | 39.62±0.00 | 61.18±0.10 |
| | ARI | 42.19±0.00 | 0.63±0.32 | 15.37±3.61 | 14.74±0.00 | 45.45±0.00 | 34.65±0.00 | 43.53±7.34 | 33.37±2.94 | 20.06±0.98 | 22.53±0.00 | 48.46±0.30 |
| | F1score | 54.26±0.00 | 30.16±2.44 | 40.04±1.96 | 31.01±0.00 | 56.91±0.00 | 47.46±0.00 | 55.81±3.44 | 46.54±2.30 | 40.42±1.60 | 43.93±0.00 | 58.51±0.25 |
| MSRCV | ACC | 78.57±0.00 | 45.05±0.57 | 68.52±1.29 | 64.29±0.00 | 70.48±0.00 | 27.62±0.00 | 46.62±0.65 | 75.29±2.81 | 58.14±2.75 | 45.50±2.72 | 84.60±0.28 |
| | NMI | 65.84±0.00 | 34.36±1.00 | 51.18±1.30 | 49.27±0.00 | 59.39±0.00 | 9.97±0.00 | 40.64±1.09 | 58.63±2.83 | 42.54±1.89 | 30.44±1.18 | 75.16±0.60 |
| | ARI | 56.66±0.00 | 20.28±0.86 | 42.01±1.78 | 41.03±0.00 | 48.87±0.00 | 3.69±0.00 | 26.50±1.03 | 52.50±3.48 | 31.99±2.56 | 19.22±1.29 | 70.05±0.54 |
| | F1score | 62.92±0.00 | 39.20±0.57 | 53.13±1.40 | 49.21±0.00 | 56.35±0.00 | 19.06±0.00 | 40.23±1.02 | 59.15±2.99 | 41.62±2.17 | 30.88±1.08 | 74.25±0.46 |
| 100leaves | ACC | 77.00±0.00 | 8.73±0.82 | 68.80±1.79 | 51.25±0.00 | 44.25±0.00 | 48.69±0.00 | 70.58±0.93 | 51.30±1.24 | 58.72±1.74 | 53.18±1.82 | 76.56±1.26 |
| | NMI | 83.41±0.00 | 32.66±2.33 | 81.15±0.61 | 68.93±0.00 | 75.27±0.00 | 77.08±0.00 | 81.38±0.34 | 70.69±0.53 | 76.63±0.71 | 73.14±0.71 | 88.30±0.45 |
| | ARI | 46.93±0.00 | 3.24±0.76 | 52.67±1.56 | 32.15±0.86 | 32.79±0.00 | 33.97±0.00 | 54.35±0.86 | 34.47±1.03 | 44.13±1.52 | 36.45±1.44 | 67.03±1.42 |
| | F1score | 47.56±0.00 | 9.99±0.56 | 58.45±1.33 | 32.79±0.00 | 33.70±0.00 | 34.87±0.00 | 59.17±0.66 | 34.47±1.03 | 44.13±1.52 | 37.11±1.42 | 67.36±1.40 |
| HW | ACC | 95.30±0.00 | 63.59±0.38 | 71.24±0.34 | 76.25±0.00 | 34.10±0.00 | 34.95±0.00 | 69.37±1.37 | 76.77±0.63 | 74.85±0.24 | 57.91±2.11 | 96.43±0.03 |
| | NMI | 90.30±0.00 | 51.73±0.10 | 56.46±0.52 | 61.10±0.00 | 24.75±0.00 | 14.90±0.00 | 58.26±0.65 | 60.76±1.37 | 54.29±0.21 | 43.45±0.84 | 91.84±0.05 |
| | ARI | 89.97±0.00 | 45.74±0.05 | 49.02±0.45 | 58.38±0.00 | 12.59±0.00 | 8.73±0.00 | 51.46±1.00 | 51.84±2.50 | 52.16±0.32 | 34.34±1.37 | 92.21±0.06 |
| | F1score | 90.96±0.00 | 53.09±0.34 | 57.57±0.51 | 62.53±0.00 | 23.01±0.00 | 18.59±0.00 | 58.38±0.58 | 56.98±2.06 | 56.93±0.29 | 40.98±1.24 | 92.99±0.06 |
| Scene | ACC | 32.69±0.00 | 27.42±1.38 | 37.77±0.68 | 33.65±0.00 | 30.35±0.00 | 30.84±0.00 | 35.59±0.48 | 32.70±1.60 | 33.96±0.01 | 25.94±0.57 | 37.52±0.87 |
| | NMI | 27.97±0.00 | 22.36±0.19 | 34.47±0.55 | 25.00±0.00 | 28.94±0.00 | 28.94±0.00 | 29.52±1.06 | 27.54±0.76 | 24.54±0.07 | 21.28±0.23 | 32.89±0.35 |
| | ARI | 12.97±0.00 | 11.28±0.87 | 19.40±0.46 | 15.36±0.00 | 14.79±0.00 | 14.82±0.00 | 18.55±0.84 | 14.74±0.62 | 14.68±0.07 | 9.69±0.21 | 18.18±0.53 |
| | F1score | 20.18±0.00 | 22.56±0.40 | 26.77±0.52 | 21.10±0.00 | 21.27±0.00 | 21.20±0.00 | 24.44±0.88 | 20.68±0.58 | 20.56±0.07 | 15.96±0.20 | 23.92±0.49 |
| EMNIST | ACC | 58.45±0.00 | 58.19±2.26 | 63.43±1.29 | 70.05±0.00 | 47.38±0.00 | 47.74±0.00 | 58.13±2.17 | 52.81±2.09 | OOM | 35.63±0.79 | 80.96±3.20 |
| | NMI | 50.65±0.00 | 42.51±0.43 | 43.23±0.96 | 51.38±0.00 | 38.72±0.00 | 42.64±0.00 | 59.26±1.65 | 36.67±0.32 | OOM | 21.03±0.70 | 77.94±1.29 |
| | ARI | 33.56±0.00 | 34.24±0.42 | 38.62±0.52 | 26.36±0.00 | 30.39±0.00 | 47.83±2.30 | 46.13±2.30 | 28.90±0.50 | OOM | 13.45±0.80 | 70.46±3.06 |
| | F1score | 41.32±0.00 | 42.58±0.29 | 45.48±0.63 | 53.05±0.00 | 34.07±0.00 | 37.58±0.00 | 53.66±1.97 | 36.11±0.44 | OOM | 22.43±0.66 | 73.48±2.69 |
| Hdigit | ACC | 67.76±0.00 | 75.68±5.89 | 78.29±4.98 | 97.62±0.00 | 62.80±0.00 | 50.65±0.00 | 81.65±4.55 | 67.61±0.90 | OOM | 54.84±1.14 | 78.80±0.05 |
| | NMI | 76.06±0.00 | 65.64±3.31 | 67.74±5.71 | 93.35±0.00 | 52.86±0.00 | 39.47±0.00 | 74.01±2.32 | 58.01±0.31 | OOM | 34.71±0.33 | 81.67±0.01 |
| | ARI | 58.19±0.00 | 62.98±5.52 | 62.55±6.30 | 94.78±0.00 | 43.47±0.00 | 29.69±0.00 | 68.76±4.81 | 51.71±0.57 | OOM | 28.86±0.59 | 73.73±0.01 |
| | F1score | 62.99±0.00 | 67.83±4.48 | 67.27±5.45 | 95.30±0.00 | 49.25±0.00 | 36.81±0.00 | 73.23±3.77 | 56.64±0.50 | OOM | 36.01±0.53 | 76.50±0.01 |

reliable and robust multi-subspace representation. Integrated into a unified framework solved via the augmented Lagrangian algorithm, TMSL can also serve as a versatile post-processing strategy to improve the performance of various existing TMVC methods.

**DSTL** (TMM 25) [48]: This method efficiently captures high-order correlations among multi-view latent semantic representations while disentangling semantic-related and unrelated components to reduce feature redundancy. By aligning semantic-related features across views through a consensus indicator, DSTL achieves scalable and robust multi-view clustering without relying on affinity graphs.

## A.6   Experimental Results with Varying Sample Alignment Ratios

To further assess the effectiveness of the proposed algorithm under different sample alignment rates, experiments were also conducted at 25% and 75% alignment rates, with detailed results shown in Tables 7-8. The findings confirm that our proposed method consistently maintains superior performance compared to other approaches. These results collectively validate the robustness and effectiveness of our algorithm in handling sample misalignment scenarios.

