# OpenReview forum: "Scalable Cross-View Sample Alignment for Multi-View Clustering with View Structure Similarity"
_NeurIPS.cc/2025/Conference — NeurIPS 2025 spotlight_

### Official Review · Reviewer_HmvP · 2025-06-25

**Clarity:** 3
**Significance:** 3
**Originality:** 4
**Rating:** 5
**Confidence:** 5

**Summary:**

A scalable multi-view clustering algorithm based on sample alignment is proposed in this paper, where an alignment matrix is learned and incorporated into a late-fusion framework to facilitate clustering in the absence of aligned samples.

**Questions:**

a. Without explicit correspondence, what principle underlies the computation of cross-view similarity between non-aligned samples?

b. What is the rationale for enforcing orthogonality on the alignment matrix M? A clear justification is currently lacking and should be provided.

**Ethical Concerns:**

["NO or VERY MINOR ethics concerns only"]

**Final Justification:**

The authors have well solved my concerns and I decide to raise the rating.

**Limitations:**

Yes

**Quality:**

4

**Strengths And Weaknesses:**

a. The manuscript is well-written and introduces a novel perspective on addressing sample misalignment in multi-view clustering.

b. Experimental results on eight benchmark datasets demonstrate the effectiveness of the proposed method.

c. It remains unclear whether cross-view similarity between unaligned representations can reliably capture semantic correspondence under strong view heterogeneity.

d. The scalability of the method to large-scale data is not thoroughly discussed, particularly regarding the computational implications and potential optimizations.

e. While the use of a k-nearest neighbor graph is intuitive, the sensitivity of the model to the choice of k warrants further analysis.

---

> ### Author Rebuttal · Authors · 2025-07-30
>
> **Q1: It remains unclear whether cross-view similarity between unaligned representations can reliably capture semantic correspondence under strong view heterogeneity.**
> **A1**: Thanks for your careful review. We would like to clarify that in our framework, the cross-view similarity between unaligned representations is not intended to directly capture semantic correspondence or high-level semantic consistency across views. Instead, it mainly estimates sample-level correspondence between views under unaligned scenarios. Specifically, our approach constructs cross-view similarity graphs by leveraging the view-specific graph structures, i.e., the similarity between the aligned samples and the unaligned samples for each view. These similarity measures are designed to identify samples that are likely to be counterparts across views, even in the presence of view heterogeneity. They are not responsible for learning semantic representations, but rather provide a relational criterion to guide the sample alignment process.
>
> **Q2: The scalability of the method to large-scale data is not thoroughly discussed, particularly regarding the computational implications and potential optimizations.**
> **A2**: Thanks for your constructive comments. As detailed in Section 4.3, the primary sources of computational cost in our proposed method include cross-sample similarity learning, cross-view similarity construction, and the late fusion stage. Among these, the largest computational complexity is the generation of the partition matrix $\mathbf{F}^v$ during the late fusion process, which leads to an overall complexity of $\mathcal{O}(n^2)$. Fortunately, this cost can be significantly reduced in practice. In particular, when more efficient or approximate strategies are adopted for generating the feature matrix $\mathbf{F}^v$, such as sparse graph construction, anchor-based clustering, or randomized low-rank approximations, the computational complexity can be reduced to nearly linear in the number of samples. Moreover, other components of the proposed method, such as CLM-based baseline view selection and cross-view similarity learning, scale linearly or near-linearly concerning the number of samples. In our experimental results, the proposed method has already demonstrated the ability to handle relatively large datasets within a reasonable time.
>
> **Q3: While the use of a k-nearest neighbor graph is intuitive, the sensitivity of the model to the choice of k warrants further analysis.**
> **A3**: Thank you for your valuable comment. We agree that the choice of the $k$-nearest neighbors parameter may impact the performance of the proposed method, and thereby we conduct additional experiments to analyze the proposed method's sensitivity to different values of $k$ under 50% sample alignment. As shown in the following experimental results, we evaluated clustering performance in terms of ACC, NMI, ARI, and F1score across a range of $k$ values ($k$=5, 9, 15, 19, 25) on six benchmark datasets. The results demonstrate that while minor performance fluctuations exist across different $k$ values, the overall performance remains relatively stable. For instance, on the HW and 100leaves datasets, the ACC and NMI remain consistently high regardless of the choice of $k$, suggesting that our method is robust to the $k$ setting. Although some datasets exhibit moderate variations, these changes do not significantly affect the general trend of the ranking of results. Moreover, the best performance does not concentrate on a single $k$ value, indicating that the model is not sensitive to a specific value. In a word, these observations confirm that our method maintains a stable and competitive performance across a reasonable range of $k$ values, suggesting that it is not highly sensitive to the neighborhood size. This robustness enhances the practicality of our method, especially when the optimal $k$ is not known a priori.
> Effect of $k$-nearest neighbors on ACC under 50% sample alignment.
> | Different $k$-nn |    Yale    |  3sources  |    MSRCV   |  100leaves |     HW     |    Scene   |
> |:----------------:|:----------:|:----------:|:----------:|:----------:|:----------:|:----------:|
> |       $k$=5      | 63.33±2.99 | 62.49±0.95 | 85.76±0.15 | 70.85±1.29 | 95.29±3.48 | 35.63±1.49 |
> |       $k$=9      | 64.79±2.48 | 61.39±0.50 | 84.45±0.32 | 70.65±1.36 | 96.07±0.04 | 36.17±0.58 |
> |      $k$=15      | 64.24±3.62 | 64.44±1.29 | 83.52±0.39 | 70.63±1.29 | 96.55±0.00 | 35.91±0.27 |
> |      $k$=19      | 64.52±3.09 | 62.57±0.33 | 83.38±0.60 | 70.76±0.98 | 95.84±2.94 | 36.60±0.75 |
> |      $k$=25      | 64.09±2.40 | 64.56±0.18 | 83.31±0.57 | 70.68±0.91 | 95.28±3.34 | 36.11±1.03 |
>
> Effect of $k$-nearest neighbors on NMI under 50% sample alignment.
> | Different $k$-nn |    Yale    |  3sources  |    MSRCV   |  100leaves |     HW     |    Scene   |
> |:----------------:|:----------:|:----------:|:----------:|:----------:|:----------:|:----------:|
> |       $k$=5      | 68.18±1.41 | 57.97±0.55 | 72.96±0.32 | 84.55±0.33 | 91.43±1.33 | 30.81±0.44 |
> |       $k$=9      | 69.34±1.16 | 57.67±0.13 | 71.66±0.45 | 84.65±0.52 | 91.42±0.07 | 29.85±0.55 |
> |      $k$=15      | 69.31±1.32 | 61.20±0.83 | 70.28±0.55 | 85.34±0.42 | 92.09±0.00 | 30.27±0.20 |
> |      $k$=19      | 69.26±1.82 | 58.81±0.31 | 70.93±0.37 | 84.50±0.41 | 91.75±1.09 | 29.53±0.68 |
> |      $k$=25      | 68.91±1.51 | 62.05±0.06 | 70.87±0.38 | 84.61±0.38 | 91.41±1.15 | 30.16±0.29 |
>
> Effect of $k$-nearest neighbors on ARI under 50% sample alignment.
> | Different $k$-nn |    Yale    |  3sources  |    MSRCV   |  100leaves |     HW     |    Scene   |
> |:----------------:|:----------:|:----------:|:----------:|:----------:|:----------:|:----------:|
> |       $k$=5      | 47.26±2.43 | 43.06±1.24 | 70.11±0.31 | 58.95±0.98 | 91.12±3.36 | 17.45±0.71 |
> |       $k$=9      | 49.23±1.67 | 41.67±0.65 | 67.97±0.58 | 59.34±1.49 | 91.48±0.08 | 16.78±0.50 |
> |      $k$=15      | 48.91±2.65 | 45.98±1.61 | 66.34±0.66 | 59.97±0.95 | 92.49±0.00 | 17.15±0.32 |
> |      $k$=19      | 49.24±2.82 | 43.88±0.32 | 66.45±0.80 | 58.94±1.14 | 91.80±2.68 | 16.91±0.63 |
> |      $k$=25      | 48.46±2.11 | 47.10±0.18 | 66.34±0.78 | 59.09±1.00 | 91.10±3.06 | 17.66±0.56 |
>
> Effect of $k$-nearest neighbors on F1score under 50% sample alignment.
> | Different $k$-nn |    Yale    |  3sources  |    MSRCV   |  100leaves |     HW     |    Scene   |
> |:----------------:|:----------:|:----------:|:----------:|:----------:|:----------:|:----------:|
> |       $k$=5      | 50.59±2.25 | 54.20±1.03 | 74.27±0.27 | 59.36±0.96 | 92.02±2.98 | 23.24±0.65 |
> |       $k$=9      | 52.44±1.56 | 53.10±0.52 | 72.43±0.50 | 59.74±1.47 | 92.33±0.07 | 22.64±0.44 |
> |      $k$=15      | 52.17±2.43 | 56.59±1.34 | 71.03±0.57 | 60.37±0.94 | 93.24±0.00 | 22.94±0.29 |
> |      $k$=19      | 52.47±2.61 | 54.82±0.27 | 71.14±0.68 | 59.34±1.12 | 92.63±2.39 | 22.78±0.56 |
> |      $k$=25      | 51.72±1.98 | 57.44±0.15 | 71.04±0.67 | 59.49±0.99 | 92.00±2.72 | 23.41±0.50 |
>
> **Q4: Without explicit correspondence, what principle underlies the computation of cross-view similarity between non-aligned samples?**
> **A4**: Thanks for your insightful comments. Our proposed method addresses this challenge through a carefully designed two-step strategy. First, within each view, every unaligned sample is represented by its affinity relative to the aligned samples in the same view, effectively embedding the unaligned data into a consistent reference criterion defined by these aligned samples. Since the aligned samples are assumed to correspond across views, cross-view similarity between unaligned samples can then be evaluated indirectly by comparing their affinities over the common aligned samples. This approach builds on the fact that non-aligned samples displaying similar affinity patterns with a common set of aligned samples across their respective views are likely to share semantic consistency, even without explicit correspondence. Furthermore, the experimental results demonstrate the effectiveness of the proposed strategies.
>
> **Q5: What is the rationale for enforcing orthogonality on the alignment matrix M? A clear justification is currently lacking and should be provided.**
> **A5**: We appreciate your insightful question. In our framework, we aim to establish soft correspondences between unaligned samples by reconstructing them based on structurally relevant information within their respective subspaces, rather than relying on hard 0-1 alignment. The orthogonality constraint is introduced to preserve the intrinsic geometric structure of the data during this reconstruction process. Specifically, orthogonality helps ensure that the transformation encoded by $\mathbf{M}$ behaves in a structure-preserving manner, avoiding distortion of sample relationships that could arise from degenerate or overly flexible alignments. This is particularly important in the presence of noise and semantic inconsistencies across views, where maintaining relative distances and semantic separability becomes challenging. Additionally, orthogonality serves as a form of regularization that stabilizes the optimization process and prevents trivial or collapsed solutions.

---

> > ### Comment · Reviewer_HmvP · 2025-08-06
> >
> > I have read the rebuttal and the concerns have been well sovled by the authors ,especially using of a k-nearest neighbor graph is intuitive, the sensitivity of the model to the choice of k warrants further analysis. Therefore, I tend to give a positive vote for this paper.

---

> > > ### Author Response · Authors · 2025-08-06
> > >
> > > Thank you very much for your positive feedback. We sincerely appreciate your time and effort in reviewing our work. We are glad to know that our responses have addressed your concerns. Your comments have been invaluable in improving the quality of our manuscript.
> > >
> > > Best regards,
> > > Authors

---

### Official Review · Reviewer_oX5B · 2025-06-26

**Clarity:** 3
**Significance:** 3
**Originality:** 4
**Rating:** 5
**Confidence:** 5

**Summary:**

A novel multi-view clustering method based on cross-view sample alignment is proposed in this manuscript, and its effectiveness is demonstrated through experiments on eight multi-view datasets.

**Questions:**

While the proposed late-fusion strategy elegantly avoids the need for early alignment, it raises an interesting question: to what extent does this design truly exploit the structural and semantic complementarities between views? One might wonder whether certain types of cross-view information—particularly those that only emerge through joint modeling—are inadvertently underutilized in the alignment process.

Another intriguing aspect is the reliance on cluster-label matching for selecting a benchmark view. This appears to work well in practice, yet it remains somewhat unclear how robust this step is to imperfections in the initial clustering. Given that early-stage clustering often involves noise or instability, especially in high-dimensional or heterogeneous views, it would be worth exploring how such variance might influence the view selection outcome—and, by extension, the final alignment quality.

**Ethical Concerns:**

["NO or VERY MINOR ethics concerns only"]

**Final Justification:**

Most of my concerns have been resolved. I would like to recommend acceptance.

**Limitations:**

Yes.

**Quality:**

3

**Strengths And Weaknesses:**

Strengths:
(1) The paper addresses the challenge of multi-view clustering with unaligned samples in a principled and practical manner. The proposed framework is conceptually clear and yields competitive results across multiple benchmarks.
(2) The use of cluster-label matching (CLM) for benchmark view selection is a noteworthy design choice. This mechanism appears to improve alignment stability, particularly in the presence of inconsistent or noisy views.
(3) The introduction of an alignment graph to connect aligned and unaligned samples enhances representational flexibility and contributes to performance gains under partial correspondence settings.

Points for Consideration:
(1) It would be helpful for the authors to clarify the specific motivations behind adopting the CLM mechanism for benchmark view selection. A brief discussion comparing it to alternative strategies (e.g., view consensus or uncertainty-based methods) could further strengthen the justification.
(2) As the alignment process is anchored on a single benchmark view, there is a potential risk that misrepresentative or noisy data within this view could influence the quality of alignment across other views. Some discussion on the mitigation of such risks would be valuable.
(3) Constructing similarity graphs between non-aligned and aligned samples is central to the proposed method. However, the paper could benefit from further analysis on how initial clustering inaccuracies might affect the propagation of alignment errors.
(4) While the alignment matrix plays a key role in sample correspondence, its robustness under noisy benchmark views remains unclear. Additional empirical or theoretical insight into the stability of alignment in such cases would enhance the completeness of the work.

---

> ### Author Rebuttal · Authors · 2025-07-30
>
> **Q1: It would be helpful for the authors to clarify the specific motivations behind adopting the CLM mechanism for benchmark view selection. A brief discussion comparing it to alternative strategies could further strengthen the justification.**
> **A1**: Thanks for your valuable suggestion. The motivation for adopting the CLM-based mechanism lies in its ability to provide a view-independent evaluation of clustering quality. Unlike many existing methods that either randomly select a baseline view or rely on view consensus, our approach aims to directly assess the intrinsic structural information of each view's clustering. Consensus-based strategies typically identify a baseline view by measuring its agreement with other views. However, such methods may favor views that are consistent yet collectively poor in quality, leading to confirmation bias across views. Similarly, uncertainty-based methods rely on the distributional confidence, but they often lack explicit consideration of clustering structure, especially in the absence of ground truth labels. In contrast, the CLM mechanism jointly evaluates intra-cluster compactness and inter-cluster separability in a scale, shift, and class cardinality-invariant manner. This allows for fair and consistent comparison across heterogeneous views. By selecting the view with the highest CLM score, the most structurally reliable view is used in the proposed method, thereby enhancing the overall robustness of cross-view alignment.
>
> **Q2: As the alignment process is anchored on a single benchmark view, there is a potential risk that misrepresentative or noisy data within this view could influence the quality of alignment across other views.**
> **A2**: Thanks for your insightful comments. We fully agree that anchoring the alignment process on a misrepresentation or noisy baseline view could propagate errors to other views and negatively affect overall performance. To mitigate this risk, our framework employs a carefully designed selection mechanism based on the CLM score, which evaluates each view's clustering structure through a combination of compactness and robustness to data scale, shift, and class imbalance. By computing the CLM score for each view, we explicitly avoid relying on random selection. Instead, we select the view with the highest structural quality as the baseline view, ensuring that it reflects well-formed clusters and minimizes the chance of incorporating noise from poorly separated structures. Furthermore, since CLM is constructed to be scale and data cardinality-invariant, it is less sensitive to outliers or imbalanced cluster sizes, making it more robust in practice. Additionally, according to the extensive experimental results in Section 5, we can observe that it consistently achieves satisfactory alignment performance across diverse multi-view datasets.
>
> **Q3: The paper could benefit from further analysis on how initial clustering inaccuracies might affect the propagation of alignment errors.**
> **A3**: We thank your insightful comments. We agree that inaccuracies in the initial clustering stages may potentially affect the construction of similarity graphs between aligned and misaligned samples, thereby influencing the propagation of alignment errors. To mitigate this, we integrate both feature-level information and structural-level information to construct the consensus representation. This dual-perspective design ensures that the model does not rely solely on any single level of information, and enables correction and compensation for localized clustering inaccuracies during the optimization of the consensus representation. Furthermore, the late fusion clustering framework is introduced to decouple the final clustering decision from the initial alignment assumptions, providing an additional layer of robustness. Essentially, even in the presence of some local misalignment, the global fusion step can improve the clustering results by weighted integration of diverse information from multiple views, without being overly affected by any single misaligned view.
>
> **Q4: While the alignment matrix plays a key role in sample correspondence, its robustness under noisy benchmark views remains unclear.**
> **A4**: We appreciate your valuable comment regarding the robustness of the learned alignment matrix under noisy baseline views. From a theoretical perspective, our alignment matrix learning framework is designed to incorporate global structural information from multiple views rather than relying solely on local pairwise similarities. This global view enables the alignment matrix to be less sensitive to localized noise or misalignment in any single baseline view. Furthermore, the alignment objective jointly optimizes for consistency across views, implicitly enforcing robustness through structural constraints and regularization. These design choices ensure that the learned correspondences capture the intrinsic relations among samples, improving stability despite noise. Empirically, we validate this theoretical robustness by conducting experiments where the alignment matrix learned by our method is applied to align clustering results from several different algorithms. Compared to using the classical Hungarian algorithm, which is primarily a local matching approach, our alignment matrix consistently achieves better clustering accuracy across diverse datasets and algorithms. The detailed experimental results are presented in the following, and the detailed introductions can refer to Section 5.7. This demonstrates that our approach can generalize beyond the specific baseline view used during alignment learning and effectively mitigate the influence of noise views. In a word, the theoretical design and empirical evidence strongly support the conclusion that our learned alignment matrix is stable and robust under the noise baseline views.
> Results of competitors on the 100leaves under a sample alignment ratio of $\rho$=50%
> | Metrics |    Setting   |   DealMVC  |    MVCAN   |    EBMGC   |    DCMVC   |    LMTC    |    TMSL    |    DSTL    |
> |:-------:|:------------:|:----------:|:----------:|:----------:|:----------:|:----------:|:----------:|:----------:|
> |   ACC   |   Unaligned  |  7.69±0.00 | 49.51±1.24 | 33.94±0.00 | 48.83±0.83 | 35.58±0.94 | 47.47±1.40 | 36.87±1.42 |
> |         | Aligned+Ours | 12.42±0.52 | 48.81±1.28 | 43.06±0.00 | 53.75±0.89 | 40.82±1.31 | 48.18±1.37 | 35.60±0.90 |
> |   NMI   |   Unaligned  | 25.34±0.44 | 69.50±0.95 | 58.22±0.00 | 67.15±0.42 | 58.75±0.55 | 69.33±0.61 | 60.53±0.55 |
> |         | Aligned+Ours | 38.01±0.46 | 69.95±0.59 | 64.52±0.00 | 72.10±0.49 | 64.58±0.80 | 70.54±0.57 | 61.09±0.53 |
> |   ARI   |   Unaligned  |  1.89±0.25 | 30.37±1.53 | 15.36±0.00 | 29.23±0.91 | 17.24±0.81 | 31.41±1.15 | 18.49±0.77 |
> |         | Aligned+Ours |  5.16±0.12 | 30.58±0.99 | 24.62±0.00 | 36.50±0.87 | 24.46±1.24 | 32.67±1.17 | 19.38±0.68 |
> | F1score |   Unaligned  |  6.68±0.15 | 37.49±1.29 | 16.16±0.00 | 35.08±0.61 | 18.07±0.81 | 32.09±1.14 | 19.35±0.75 |
> |         | Aligned+Ours | 11.47±0.14 | 37.74±0.93 | 25.33±0.00 | 41.38±0.85 | 25.21±1.23 | 33.34±1.15 | 20.23±0.67 |
>
> **Q5: To what extent does this design truly exploit the structural and semantic complementarities between views?**
> **A5**: Thanks for your constructive comments. Indeed, it is critical to ensure that the late-fusion strategy fully leverages the structural and semantic information among views. To address this, we designed our framework with the following considerations. Firstly, while the fusion process is conducted at a later stage, we do not rely solely on independent view-wise clustering results. Instead, the alignment phase incorporates both feature-level information and graph-structural similarities, which are used to learn a unified latent representation space. This design allows the alignment process to indirectly capture inter-view dependencies before fusion. Secondly, to address the potential underutilization of cross-view information that only emerges through joint modeling, we introduce a weighted consensus mechanism in the fusion step. Specifically, each view contributes to the final clustering based on its alignment consistency and structural reliability, allowing high-quality views to reinforce low-quality ones and flexibly preserving multi-view complementary information. In summary, although our method does not perform explicit joint modeling in the traditional sense, it captures structural and semantic information through a combination of unified representation learning and adaptive weighted fusion, and the experimental results demonstrate the effectiveness of our proposed method.
>
> **Q6: Another intriguing aspect is the reliance on cluster-label matching for selecting a benchmark view.**
> **A6**: We sincerely appreciate your insightful comments. Unlike methods that assess each view independently, CLM measures the relative consistency of each view concerning all other views through a pairwise, normalized label-matching function. By aggregating over all $\binom{k}{2}$ cluster pairs, CLM captures global structural similarity in a redundant and smoothed manner, which effectively reduces the impact of local noise or instability in individual views. Furthermore, this strategy enables CLM to select the view that is most structurally aligned with the consensus of the remaining views, even when all views contain some level of noise. As such, the view selection process becomes inherently more robust than strategies that either randomly choose a baseline view or rely on the absolute clustering performance of a single view. Furthermore, as observed in Section 5.7, using our alignment matrix to align the clustering results of other algorithms achieves superior performance compared to the traditional Hungarian matching approach, which demonstrates the robustness and generalization of the selected baseline view.

---

> > ### Comment · Reviewer_oX5B · 2025-08-06
> >
> > Thank you for your detailed response. Most of my concerns have been thoroughly addressed in the authors' response, including clarifications on CLM selection, robustness analysis, and view complementarity. So I decided to raise my rating.

---

> > > ### Author Response · Authors · 2025-08-06
> > >
> > > We sincerely appreciate your thoughtful comments and your decision to raise the score. Your recognition is truly encouraging and valuable to us.
> > >
> > > Best regards,
> > > Authors

---

### Official Review · Reviewer_gq64 · 2025-06-28

**Clarity:** 2
**Significance:** 3
**Originality:** 3
**Rating:** 4
**Confidence:** 4

**Summary:**

The proposed method addresses unalignment samples in multi-view clustering by aligning views through a benchmark-based cluster-label matching strategy and constructing cross-view similarity graphs, enabling effective clustering without requiring sample correspondence.

**Questions:**

1. Can the proposed method be effectively applied to cases where samples are entirely unaligned across views?

2. The same approach is used for both unaligned sample representation and cross-view similarity—are these the only feasible choices, or can alternative methods be employed?

3. As alignment performance depends on the benchmark view, how is its reliability ensured?

4. While the datasets are standard in multi-view clustering, could the authors clarify how they were preprocessed to simulate sample misalignment?

5. The proposed method belongs to MVC methods with shallow models, so it is recommended to compare more SOTA shallow MVC methods like tensor-based or subspace-based methods.

**Ethical Concerns:**

["NO or VERY MINOR ethics concerns only"]

**Final Justification:**

Thanks for authors' response. This paper proposed an interesting method with promising results, and I have no further questions.

**Limitations:**

Yes.

**Paper Formatting Concerns:**

N/A.

**Quality:**

3

**Strengths And Weaknesses:**

The strengths of this manuscript are listed as follows:

The proposed method effectively handles unalignment multi-view data by eliminating the need for strict sample correspondence across views; The use of cluster-label matching to select a benchmark view ensures more reliable and consistent alignment across different views; By integrating the alignment criterion into a late-fusion framework, SSA-MVC maintains scalability and flexibility across diverse datasets.

The weaknesses of this manuscript are listed as follows:

Can the proposed method be effectively applied to cases where samples are entirely unaligned across views; the same approach is used for both unaligned sample representation and cross-view similarity—are these the only feasible choices, or can alternative methods be employed; as alignment performance depends on the benchmark view, how is its reliability ensured; while the datasets are standard in multi-view clustering, could the authors clarify how they were preprocessed to simulate sample misalignment?

---

> ### Author Rebuttal · Authors · 2025-07-30
>
> **Q1: Can the proposed method be effectively applied to cases where samples are entirely unaligned across views?**
> **A1**: Thanks for your valuable comment. Our method remains effective even in the absence of any sample-level correspondence across views. Rather than depending on explicit alignment, it captures the shared clustering structure implicitly present across different modalities. This is achieved by projecting data into a latent space where structural patterns are preserved through orthogonality constraints. These constraints ensure that the alignment process respects the intrinsic geometry of each view, enabling the model to discover coherent groupings without relying on direct sample matching. In fully unaligned scenarios (i.e., alignment ratio $\rho$=0), we adapt our strategy by discarding alignment-based similarity measures and instead computing cross-view relationships via the affinity between view-specific partition matrices. This allows the model to emphasize clustering-level consistency, which serves as a reliable proxy for alignment under severe view discrepancy. To assess the practical effectiveness of our method in such settings, we conduct comparisons with two representative models designed for unaligned multi-view clustering: VSC_mH and OpVuC. Extensive experiments on six benchmark datasets demonstrate that our method consistently achieves higher performance across most metrics. These results confirm that even in the most challenging case of complete sample misalignment, our framework successfully exploits latent structural information to produce robust clustering results.
> Clustering results under fully unaligned views
> |  Metric | Methods |    Yale    |  3sources  |    MSRCV   |  100leaves |     HW     |    Scene   |
> |:-------:|:-------:|:----------:|:----------:|:----------:|:----------:|:----------:|:----------:|
> |         |  Vsc_mH | 36.36±0.00 | 40.24±0.00 | 39.05±0.00 | 13.94±0.00 | 13.50±0.00 | 29.45±0.00 |
> |   ACC   |  OpVuC  | 24.24±0.00 | 24.26±0.00 | 24.29±0.00 |  8.75±0.00 | 12.90±0.00 | 10.90±0.00 |
> |         |   Ours  | 59.36±2.40 | 55.77±0.42 | 68.31±0.66 | 54.93±1.37 | 19.25±0.95 | 19.88±0.65 |
> |         |  Vsc_mH | 41.79±0.00 | 11.27±0.00 | 21.78±0.00 | 44.50±0.00 |  3.42±0.00 | 27.85±0.00 |
> |   NMI   |  OpVuC  | 28.96±0.00 |  4.53±0.00 |  5.97±0.00 | 32.91±0.00 |  0.93±0.00 |  3.52±0.00 |
> |         |   Ours  | 63.97±1.26 | 50.21±0.97 | 49.65±0.93 | 73.11±0.86 |  5.11±0.36 |  9.98±0.63 |
> |         |  Vsc_mH | 14.61±0.00 | 11.07±0.00 | 12.71±0.00 |  4.82±0.00 |  0.10±0.00 | 14.39±0.00 |
> |   ARI   |  OpVuC  |  2.57±0.00 |  0.27±0.00 |  1.59±0.00 |  1.08±0.00 |  0.02±0.00 |  0.56±0.00 |
> |         |   Ours  | 40.97±1.94 | 37.35±0.57 | 43.51±1.02 | 39.29±1.28 |  2.25±0.23 |  4.78±0.34 |
> |         |  Vsc_mH | 20.49±0.00 | 31.99±0.00 | 25.05±0.00 |  6.09±0.00 | 15.99±0.00 | 21.10±0.00 |
> | F1score |  OpVuC  |  8.74±0.00 | 19.24±0.00 | 17.57±0.00 |  2.53±0.00 | 10.97±0.00 |  9.06±0.00 |
> |         |   Ours  | 44.75±1.80 | 49.59±0.46 | 51.44±0.88 | 39.90±1.26 | 12.06±0.21 | 11.34±0.32 |
>
> **Q2: The same approach is used for both unaligned sample representation and cross-view similarity—are these the only feasible choices, or can alternative methods be employed?**
> **A2**: Thank you for your insightful comment. In our method, we adopt the adaptive neighbor graph learning for both unaligned sample representation learning and cross-view similarity learning. This approach provides a unified and effective way to capture local geometric structures and inter-view relationships. Of course, we acknowledge that this is not the only feasible solution. Alternative strategies, such as metric learning techniques, multiple kernel learning, could also be employed for them. We chose the current method due to its flexibility and effectiveness in our preliminary studies. Therefore, future work could explore the impact of these alternative methods.
>
> **Q3: As alignment performance depends on the benchmark view, how is its reliability ensured?**
> **A3**: Thanks for your constructive comments. In the existing alignment-based multi-view clustering methods, the baseline view is either randomly selected or chosen heuristically without considering the intrinsic clustering quality of each view. This randomness can lead to sub-optimal alignment, especially when the selected view exhibits poor cluster structure or noise. To address this, we introduced a baseline view selection mechanism based on the CLM algorithm, and the detailed introductions can refer to Section 3.2. As presented in Eqs. (4)-(5), the measures assess the separation and compactness of cluster structures in a scale-invariant, shift-invariant, and class-size-invariant manner, enabling a fair comparison of clustering quality across unaligned views. Specifically, we compute the CLM score for each view and select the one with the highest score as the baseline view, thereby ensuring that alignment is guided by the most reliable structural information. Therefore, compared with the existing baseline view selection methods, the proposed method mitigates the limitations of random selection and enhances the stability and performance of the alignment process.
>
> **Q4: While the datasets are standard in multi-view clustering, could the authors clarify how they were preprocessed to simulate sample misalignment?**
> **A4**: Thank you for your valuable comments. As existing multi-view datasets are originally aligned across views, we simulate the sample misalignment scenario by artificially disrupting the arrangement of the samples. Specifically, we first apply the CLM algorithm to determine the baseline view. Then, we select a fixed proportion of samples with the same indices across views to serve as aligned samples for each view. The remaining samples in each view are considered as unaligned by randomly shuffling their order. Additionally, in the baseline view, we preserve the original sample order to maintain consistency with the ground-truth labels. Through this process, we construct multi-view datasets with varying degrees of sample misalignment, which allow us to effectively evaluate the robustness of our proposed method under sample misalignment scenarios.
>
> **Q5: It is recommended to compare more SOTA shallow MVC methods like tensor-based or subspace-based methods.**
> **A5**: Thank you for your insightful suggestion. In our experimental comparison, we have added two representative shallow multi-view clustering methods, i.e., TPCH [1] and ESTMC [2], and we report results under a 50% sample alignment ratio, with and without Hungarian alignment. The results demonstrate that our proposed method consistently outperforms them across all datasets in four clustering metrics. Notably, applying Hungarian alignment to these baselines does not necessarily lead to performance gains and sometimes even degrades the results, indicating that rigid post-processing alignment may be unreliable when sample correspondences are uncertain. In contrast, our method achieves significantly better and more stable performance by jointly modeling soft sample alignment and cross-view structural consistency, further demonstrating its effectiveness in handling partially aligned multi-view data.
> Clustering performance under a sample alignment ratio $\rho$=50%
> | Metrics |     Methods     |    Yale    |  3sources  |    MSRCV   |  100leaves |     HW     |    Scene   |
> |:-------:|:---------------:|:----------:|:----------:|:----------:|:----------:|:----------:|:----------:|
> |         |       TPCH      | 27.88±0.00 | 30.77±0.00 | 33.81±0.00 | 32.38±0.00 | 25.20±0.00 | 19.15±0.00 |
> |         |  TPCH+Hungarian | 33.94±0.00 | 30.18±0.00 | 27.62±0.00 | 16.25±0.00 | 17.95±0.00 | 15.43±0.00 |
> |   ACC   |      ESTMC      | 51.42±3.74 | 49.59±0.91 | 54.95±2.22 | 35.53±1.04 | 69.63±4.64 | 23.81±0.92 |
> |         | ESTMC+Hungarian | 32.94±1.71 | 36.92±0.56 | 45.21±1.96 | 35.54±1.18 | 45.94±0.81 | 20.77±0.50 |
> |         |       Ours      | 64.24±3.62 | 64.44±1.29 | 83.52±0.39 | 70.63±1.29 | 96.55±0.00 | 35.91±0.27 |
> |         |       TPCH      | 32.65±0.00 |  7.03±0.00 | 14.31±0.00 | 56.35±0.00 |  9.51±0.00 | 12.65±0.00 |
> |         |  TPCH+Hungarian | 35.95±0.00 |  8.31±0.00 |  8.18±0.00 | 45.32±0.00 |  3.17±0.00 |  5.27±0.00 |
> |   NMI   |      ESTMC      | 56.44±3.05 | 34.70±1.56 | 45.57±2.96 | 58.76±0.63 | 61.21±2.06 | 17.37±0.78 |
> |         | ESTMC+Hungarian | 35.61±1.59 | 19.24±0.73 | 23.58±1.52 | 58.84±0.73 | 23.72±0.55 | 14.11±0.31 |
> |         |       Ours      | 69.31±1.32 | 61.20±0.83 | 70.28±0.55 | 85.34±0.42 | 92.09±0.00 | 30.27±0.20 |
> |         |       TPCH      |  5.33±0.00 |  2.54±0.00 |  6.72±0.00 | 12.62±0.00 |  5.14±0.00 |  5.78±0.00 |
> |         |  TPCH+Hungarian |  8.84±0.00 |  3.41±0.00 |  2.08±0.00 |  3.10±0.00 |  1.30±0.00 |  2.93±0.00 |
> |   ARI   |      ESTMC      | 30.82±4.14 | 25.26±1.08 | 32.04±2.42 | 17.15±0.91 | 52.12±3.56 |  7.97±0.41 |
> |         | ESTMC+Hungarian |  8.19±1.26 |  9.98±1.00 | 15.50±1.07 | 17.24±1.03 | 18.60±0.48 |  6.49±0.23 |
> |         |       Ours      | 48.91±2.65 | 45.98±1.61 | 66.34±0.66 | 59.97±0.95 | 92.49±0.00 | 17.15±0.32 |
> |         |       TPCH      | 11.39±0.00 | 22.24±0.00 | 19.87±0.00 | 13.66±0.00 | 14.74±0.00 | 12.33±0.00 |
> |         |  TPCH+Hungarian | 14.60±0.00 | 21.95±0.00 | 16.32±0.00 |  4.12±0.00 | 12.73±0.00 | 10.53±0.00 |
> | F1score |      ESTMC      | 35.18±3.89 | 39.76±0.87 | 41.53±2.05 | 17.98±0.90 | 56.94±3.18 | 14.34±0.39 |
> |         | ESTMC+Hungarian | 13.96±1.18 | 27.77±0.65 | 27.33±0.86 | 18.08±1.02 | 26.85±0.44 | 12.91±0.22 |
> |         |       Ours      | 52.17±2.43 | 56.59±1.34 | 71.03±0.57 | 60.37±0.94 | 93.24±0.00 | 22.94±0.29 |
>
> [1] Wang Z, Li X, Sun Y, et al. TPCH: tensor-interacted projection and cooperative hashing for multi-view clustering, AAAI. 2025.
> [2] Ji J, Feng S. Anchors crash tensor: efficient and scalable tensorial multi-view subspace clustering, IEEE TPAMI, 2025.

---

> > ### Comment · Reviewer_gq64 · 2025-08-06
> >
> > Thanks for authors' response. This paper proposed an interesting method with promising results, and I have no further questions.

---

> > > ### Author Response · Authors · 2025-08-06
> > >
> > > Thank you for your positive response. We appreciate your time and valuable feedback, which have greatly helped improve our work.
> > >
> > > Best regards,
> > > Authors

---

### Official Review · Reviewer_PvMp · 2025-07-02

**Clarity:** 3
**Significance:** 3
**Originality:** 3
**Rating:** 4
**Confidence:** 4

**Summary:**

This paper proposes a scalable sample-alignment-based multi-view clustering method. The proposed approach first employs a cluster-label matching algorithm to select the view, and then constructs representations of non-aligned samples by computing their similarities with aligned samples. Extensive experiments demonstrate the effectiveness of the proposed method.

**Questions:**

(1) The title of this paper is 'Scalable Cross-View Sample Alignment for Multi-View Clustering with View Structure Similarity'. However, it lacks the most important experiments with fully misaligned views. This raises a question: Is the proposed method unable to address the issue of fully misaligned views?

(2) This paper considers the issue of view misalignment. Therefore, performance validation should be compared more with the SOTA methods of view misalignment instead of traditional multi-view clustering methods.

(3) The proposed method uses a feature rotation matrix to align the feature space. Is the rotation matrix effective for the sample, i.e., can the learned rotation matrix be used to output aligned samples?

(4) The presentation of the paper can be further improved. For example, the roles of each component in Eqs. (4), (5) and (7) should be specifically explained.

**Ethical Concerns:**

["NO or VERY MINOR ethics concerns only"]

**Final Justification:**

After reading the other reviewers' comments and the authors' response, most of my concers have been addressed. So I raise the rating from 3 to 4.

**Limitations:**

Yes

**Quality:**

3

**Strengths And Weaknesses:**

Strengths:
(1) This paper proposes a novel alignment strategy to address the issue with partially aligned views. Extensive experiments demonstrate the effectiveness of the proposed method.

Weaknesses:
(1) The title of this paper is 'Scalable Cross-View Sample Alignment for Multi-View Clustering with View Structure Similarity'. However, it lacks the most important experiments with fully misaligned views. This raises a question: Is the proposed method unable to address the issue of fully misaligned views?

(2) This paper considers the issue of view misalignment. Therefore, performance validation should be compared more with the SOTA methods of view misalignment instead of traditional multi-view clustering methods.

(3) The proposed method uses a feature rotation matrix to align the feature space. Is the rotation matrix effective for the sample, i.e., can the learned rotation matrix be used to output aligned samples?

(4) The writing of this paper can be further improved. For example, the roles of each component in Eqs. (4), (5) and (7) should be specifically explained.

---

> ### Author Rebuttal · Authors · 2025-07-30
>
> **Q1: Is the proposed method unable to address the issue of fully misaligned views?**
> **A1**: Thanks for your valuable comment. Although our method is primarily designed for partially aligned multi-view scenarios, it is also theoretically and practically applicable to fully unaligned views. Theoretically, our framework is built on the assumption that all views share a common underlying clustering structure. Even in the absence of explicit sample correspondence, samples belonging to the same cluster across different views exhibit implicit structural consistency. To capture this, we employ an orthogonality-constrained alignment matrix $\mathbf{M}$ that aligns samples based on their structural behavior in the latent space, rather than their correspondence. In practice, under fully unaligned scenarios, i.e., alignment ratio $\rho=0$, we adapt our strategy by discarding the alignment-based similarity construction. Instead, we compute cross-view similarity based on the affinity between partition matrices $\mathbf{F}^v$ of different views. This allows us to preserve and exploit high-level clustering consistency without relying on direct sample-wise alignment. The learned alignment relationship thus enables indirect alignment by capturing common clustering structures. Furthermore, to demonstrate the effectiveness of our proposed method on fully unaligned scenarios, we compare it with Vsc_mH and OpVuC, which are among the few existing methods capable of handling fully unaligned multi-view data. According to the experimental results, we can find that our method outperforms the competitors in most of the datasets. Thus, these results confirm the adaptability of our method in addressing full sample misalignment by leveraging shared structural information at the partition level.
> Clustering results of compared methods under fully unaligned views (alignment ratio $\rho=0$)
> |  Metric | Methods |    Yale    |  3sources  |    MSRCV   |  100leaves |     HW     |    Scene   |
> |:-------:|:-------:|:----------:|:----------:|:----------:|:----------:|:----------:|:----------:|
> |         |  Vsc_mH | 36.36±0.00 | 40.24±0.00 | 39.05±0.00 | 13.94±0.00 | 13.50±0.00 | 29.45±0.00 |
> |   ACC   |  OpVuC  | 24.24±0.00 | 24.26±0.00 | 24.29±0.00 |  8.75±0.00 | 12.90±0.00 | 10.90±0.00 |
> |         |   Ours  | 59.36±2.40 | 55.77±0.42 | 68.31±0.66 | 54.93±1.37 | 19.25±0.95 | 19.88±0.65 |
> |         |  Vsc_mH | 41.79±0.00 | 11.27±0.00 | 21.78±0.00 | 44.50±0.00 |  3.42±0.00 | 27.85±0.00 |
> |   NMI   |  OpVuC  | 28.96±0.00 |  4.53±0.00 |  5.97±0.00 | 32.91±0.00 |  0.93±0.00 |  3.52±0.00 |
> |         |   Ours  | 63.97±1.26 | 50.21±0.97 | 49.65±0.93 | 73.11±0.86 |  5.11±0.36 |  9.98±0.63 |
> |         |  Vsc_mH | 14.61±0.00 | 11.07±0.00 | 12.71±0.00 |  4.82±0.00 |  0.10±0.00 | 14.39±0.00 |
> |   ARI   |  OpVuC  |  2.57±0.00 |  0.27±0.00 |  1.59±0.00 |  1.08±0.00 |  0.02±0.00 |  0.56±0.00 |
> |         |   Ours  | 40.97±1.94 | 37.35±0.57 | 43.51±1.02 | 39.29±1.28 |  2.25±0.23 |  4.78±0.34 |
> |         |  Vsc_mH | 20.49±0.00 | 31.99±0.00 | 25.05±0.00 |  6.09±0.00 | 15.99±0.00 | 21.10±0.00 |
> | F1score |  OpVuC  |  8.74±0.00 | 19.24±0.00 | 17.57±0.00 |  2.53±0.00 | 10.97±0.00 |  9.06±0.00 |
> |         |   Ours  | 44.75±1.80 | 49.59±0.46 | 51.44±0.88 | 39.90±1.26 | 12.06±0.21 | 11.34±0.32 |
>
> **Q2: Should be compared more with the SOTA methods of view misalignment instead of traditional multi-view clustering methods.**
> **A2**: Thank you for your valuable suggestion. To better evaluate the effectiveness of our method in handling view misalignment, we have included tow representative state-of-the-art methods specifically designed for unaligned multi-view clustering, i.e., TUMCR[1] and DAGF[2], and the corresponding experimental results are presented in the following, where we report ACC under two different alignment ratios: 0\%(fully unaligned) and 50\% (partially aligned). As shown, our method consistently achieves superior performance across six benchmark datasets in both settings. For example, under the fully unaligned scenario, our method outperforms TUMCR and DAGF by large margins on datasets such as MSRCV and 100leaves. Similar trends can be observed under the other datasets. Therefore, these results demonstrate the adaptability of our method in dealing with both fully and partially misaligned scenarios.
> ACC comparison with view misalignment multi-view clustering methods under different alignment ratios
> | Alignment Ratios | Methods |    Yale    |  3sources  |    MSRCV   |  100leaves |     HW     |    Scene   |
> |:----------------:|:-------:|:----------:|:----------:|:----------:|:----------:|:----------:|:----------:|
> |                  |  TUMCR  | 29.09±0.00 | 36.09±0.00 | 26.67±0.00 | 47.44±0.00 | 15.70±0.00 |  9.59±0.00 |
> |        0%        |   DAGF  | 17.33±0.24 | 33.31±0.00 | 26.05±1.45 |  8.61±0.08 | 18.02±2.34 | 11.83±0.56 |
> |                  |   Ours  | 59.36±2.40 | 55.77±0.42 | 68.31±0.66 | 54.93±1.37 | 19.25±0.95 | 19.88±0.65 |
> |                  |  TUMCR  | 35.76±0.00 | 42.60±0.00 | 37.14±0.00 | 41.75±0.00 | 41.35±0.00 | 14.51±0.00 |
> |        50%       |   DAGF  | 18.36±0.36 | 33.90±0.01 | 27.90±1.37 |  9.46±0.11 | 22.37±2.53 | 13.44±0.57 |
> |                  |   Ours  | 64.24±3.62 | 64.44±1.29 | 83.52±0.39 | 70.63±1.29 | 96.55±0.00 | 35.91±0.27 |
>
> [1] Ji J, Feng S, Li Y. Tensorized unaligned multi-view clustering with multi-scale representation learning[C]// KDD. 2024.
> [2] Jiang H, Tao H, Jiang Z, et al. Unaligned multi-view clustering via diversified anchor graph fusion[J]. Pattern Recognition, 2025.
>
> **Q3: Can the learned rotation matrix be used to output aligned samples?**
> **A3**: Thanks for your constructive comments. In our method, the feature rotation matrix $\mathbf{R}$ is primarily learned in the latent feature space to align the view-specific representations with the unified representation, rather than to directly realign the original samples. Specifically, $\mathbf{M}$ operates on the extracted features of samples, mapping them into a shared feature space. Therefore, while the learned rotation matrix can produce aligned feature representations, it does not yield aligned samples.
>
> **Q4: The roles of each component in Eqs. (4), (5), and (7) should be specifically explained.**
> **A4**: Thanks for your advice. The metric in Eq.(4) acts as an enhanced separation-compactness ratio that evaluates the structural quality of a clustering $\mathbf{Y}$ with respect to the entire dataset $\mathbf{X}$. Specifically, H integrates scale-invariant normalization through $\sigma_{d^2}$, exponential smoothing to mitigate the effects of data shifts, and a balanced weighting of inter-cluster and intra-cluster distances. As a result, H provides a robust and comparable measure of cluster quality across different views, even when the data distributions and cluster cardinalities vary. On the basis of Eq.(4), Eq.(5) defines the final clustering quality measure, i.e., it averages pairwise cluster scores and applies a logistic transformation to normalize the range to 0-1. This design not only ensures class-cardinality invariance but also provides a directly interpretable and bounded quality score. Therefore, with Eqs. (4)-(5), we can quantify both intra-view compactness and inter-view separability in a scale-free manner and consistently identify the view that fully preserves the intrinsic structure of the data, regardless of sample size, feature scales, or the number of clusters. In this way, we can avoid biases caused by poorly aligned or low-quality views for subsequent cross-view alignment.
> After selecting the base view according to the Eqs.(4)-(5), the final clustering objective function can be formulated, as presented in Eq.(7). The first part of Eq.(7) is a sample alignment-based late fusion clustering framework, where the sample alignment matrices $\mathbf{M}^v$ and the feature rotation matrices $\mathbf{R}^v$ are jointly learned to map each view-specific clustering result $\mathbf{F}^v$ into a shared space, enabling the consensus partition $\mathbf{F}^*$ to integrate multi-view information effectively. The second part introduces a structure-guided alignment term, where $\mathbf{S}^v$ encodes the similarity between the non-aligned samples of each view and the baseline view. This regularization term drives $\mathbf{M}^v$ to capture the correct correspondence by preserving cross-view structural relationships.

---

> > ### Comment · Reviewer_PvMp · 2025-08-05
> >
> > I thank the authors for providing a comprehensive response to my comments. The response addresses most of my concerns. After reading the rebuttal and the other reviewers' comments, I would like to raise my rating.

---

> ### Author Response · Authors · 2025-08-05
>
> Thank you for your kind response. We are delighted to hear that our rebuttal has addressed your concerns. We sincerely appreciate your thoughtful comments and the time you devoted to reviewing our manuscript. Your constructive feedback has greatly helped us improve the quality of our work. We are also grateful for your positive evaluation and the final recommendation to raise the score.
>
> Best regards,
>
> The Authors

---

### Note · Authors · 2025-08-11

**Dear SACs, ACs, and Reviewers,**
We sincerely appreciate your time and effort in reviewing our paper and providing constructive feedback. We are grateful for the reviewers’ recognition of the novelty and contributions of our work.
Below, we present a summary addressing several common concerns raised by the reviewers:
**1. Applicability to fully unaligned scenarios**
Although our proposed method is primarily designed for partially unaligned multi-view data, it is also applicable to fully unaligned scenarios. The key difference is that, instead of relying on the original feature representations of unaligned samples to construct cross-view similarities, we construct them from the partition matrices of each view. This approach enables us to capture and preserve the clustering structure consistency across views without requiring a strict one-to-one sample correspondence. Once the cross-view similarity graph is constructed, our overall objective function can be applied directly to complete the clustering task.
**2. Reliability in selecting the baseline view**
Unlike existing methods that select a baseline view randomly or via heuristic search, our method rigorously evaluates and scores the initial quality of each view in a scale-invariant, shift-invariant, and class-size-invariant manner. By selecting the view with the highest overall score as the baseline view, we reduce the risk of degraded performance caused by a low-quality reference view.
**3. Advancement of performance**
Beyond comparisons with ten state-of-the-art multi-view clustering methods, we further compare our proposed method with two unaligned multi-view clustering algorithms, a subspace clustering method, and a tensor-based multi-view clustering method. The results consistently demonstrate the effectiveness and superiority of our proposed method.
Thank you again for your valuable feedback.
Best regards,
The Authors

---

### Decision · Program_Chairs · 2025-09-17

**Decision:**

Accept (spotlight)

**Comment:**

This paper proposes a scalable sample-alignment-based multi-view clustering (MVC) method to address the challenge of partially aligned views. The key innovation lies in a cluster-label matching (CLM) strategy to select a benchmark view, followed by constructing cross-view similarity graphs to represent non-aligned samples. The method integrates alignment into a late-fusion framework, enabling clustering without strict sample correspondence. Experiments on eight benchmarks demonstrate its effectiveness, particularly in partial alignment scenarios. Reviewers highlight the novelty of the alignment strategy, its scalability, and competitive performance against traditional MVC methods. Combining their consensus acceptance recommendation after rebuttal, the paper is recommended to be accepted.